# The Microscopic Mechanisms Involved in Superexchange

## Jacques Curély †

Laboratoire Ondes et Matière d'Aquitaine, UMR 5798, University of Bordeaux, 351 Cours de la Libération, CEDEX, 33405 Talence, France; jacques.curely@u-bordeaux.fr
† Dedicated to Professor Roland Georges.

**Abstract:** In earlier work, we previously established a formalism that allows to express the exchange energy $J$ vs. fundamental molecular integrals without crystal field, for a fragment A–X–B, where A and B are $3d^1$ ions and X is a closed-shell diamagnetic ligand. In this article, we recall this formalism and give a physical interpretation: we may rigorously predict the ferromagnetic ($J < 0$) or antiferromagnetic ($J > 0$) character of the isotropic (Heisenberg) spin-spin exchange coupling. We generalize our results to $nd^m$ ions ($3 \leq n \leq 5$, $1 \leq m \leq 10$). By introducing a crystal field we show that, starting from an isotropic (Heisenberg) exchange coupling when there is no crystal field, the appearance of a crystal field induces an anisotropy of exchange coupling, thus leading to a $z$-$z$ (Ising-like) coupling or a $x$-$y$ one. Finally, we discuss the effects of a weak crystal field magnitude ($3d$ ions) compared to a stronger ($4d$ ions) and even stronger one ($5d$ ions). In the last step, we are then able to write the corresponding Hamiltonian exchange as a spin-spin one.

**Keywords:** superexchange; magnetic orbital; Anderson model; isotropic and anisotropic spin-spin exchange couplings

## 1. Introduction

It was necessary to wait until the end of the 1950s to achieve a good understanding of superexchange interactions, when Anderson first proposed the theory of coupling between identical ions, characterized by a $3d^m$ electronic configuration without orbital degeneracy ($m = 1$) [1,2], later generalized to $m > 1$. In this case, the exchange Hamiltonian is of the Heisenberg–Dirac type: $J\boldsymbol{s}_1.\boldsymbol{s}_2$. Anderson's publication has become the starting point for generalizations, notably with the introduction of orbital degeneracy [2–9].

We have proposed a first generalization of the various mechanisms involved in superexchange for identical $3d^1$ ions: the exchange energy constant $J$ has been expressed vs. fundamental molecular integrals, characterizing each of the σ–type bonds created by the presence of a diamagnetic ligand and a similar (or different) magnetic ion [10].

*What is new in this paper?* So far, no physical interpretation has been brought. In this article, we give the physical interpretation, and we generalize to ions $nd^m$ ($3 \leq n \leq 5$, $1 \leq m \leq 10$). In addition, we rigorously show for the first time that the Hamiltonian is given by $J\boldsymbol{s}_i.\boldsymbol{s}_j$, where $J$ is the "exchange constant". The key physical points are as follows:

- $J$ is expressed vs. fundamental molecular integrals in the absence of a crystal field, uniquely, for the sake of simplicity; we show that the introduction of a crystal field may be achieved very easily, thus allowing us to discuss further the notion of anisotropic couplings;
- For the first time, we may rigorously predict the ferromagnetic ($J < 0$) or antiferromagnetic ($J > 0$) character of spin-spin couplings whereas, so far, we have dealt with empirical rules, i.e., the Goodenough–Kanamori rules published between the middle of the 1950s and the beginning of the 1960s [11–14].

At that time, the magnetic compounds of interest were essentially the oxides and fluorides of transition elements, as well as "natural" garnets, notably ferrites. The Goodenough–Kanamori rules gave satisfactory results regarding the sign of $J$ but there were some failures.

Indeed, although these rules can predict ferromagnetic and antiferromagnetic arrangements, they do not give the magnitude of *J*. Experimental works have pointed out that superexchange could also lead to ferromagnetic spin arrangements but with a magnitude of *J* not predicted by Anderson's model, which mainly predicted antiferromagnetic arrangements [11]. As a result, the first improvements to Anderson's model have consisted of taking into account polarization effects [3], thus leading researchers to introduce perturbation expansions of the initial model; however, this turned out to be insufficient.

In addition, at the beginning of the 1990s, new magnetic compounds that did not exist in nature were synthetized [15–18]. These "synthetized" compounds were in opposition to the "natural" ones encountered until they were characterized by the introduction of organic ligands of variable length between magnetic cations. Their introduction deeply undermined the previous interpretation of the *J* sign.

This is the reason for which:

- Here, without a crystal field, we deal with a theoretical model, from which we derive the conclusion that when coulombic interactions are dominant, our model follows Hund's rule and we explain why couplings are automatically ferromagnetic; when coulombic interactions are no longer dominant, our model is equivalent to the molecular orbital one and couplings are always antiferromagnetic (except in a particular case, where couplings are ferromagnetic but present a small absolute value of *J*);
- By introducing the notion of a crystal field, we discuss how passing from an isotropic (Heisenberg) coupling to an anisotropic one (*z-z*, i.e., an Ising-like coupling or an *x-y* one); in addition, from the theoretical expression of *J*, we may also predict the ferromagnetic (*J* < 0) or antiferromagnetic (*J* > 0) character of spin-spin couplings as in the absence of a crystal field, which is the key finding of this article.

In addition, in each case, the model allows us to express the magnitude of *J* for any sign. *In this review article, we do not consider the polarization effect of the involved bonds.*

As a result, the paper is ordered as follows. Section 2 is devoted to the microscopic mechanisms involved in superexchange. We consider the most general case of *two different magnetic* $3d^1$ *ions,* A and B, *characterized by σ–type bonds on each side of the diamagnetic bridge,* X. *The cationic orbitals are of a d-type for* A *and* B, *whereas that of the diamagnetic ligand* X *is of the s- or p-type.* We notably define and justify the general assumptions used for developing our theoretical model.

Under these conditions, we show that this general treatment allows the researcher to calculate superexchange interaction within the isolated fragment A–X–B through the construction of the intermediate cationic states, which leads to the determination of the collective state. The matrix associated with the corresponding Hamiltonian is logically derived, as well as the energy spectrum. By doing so, the corresponding exchange energy *J* is expressed vs. the key molecular integrals characterizing the two σ bonds of A–X–B.

In Section 3, we give a full closed-form expression for *J* vs. key molecular integrals as well as physical interpretations. As a result, it becomes possible to predict and interpret the sign of *J* as well as its magnitude.

Finally, we introduce the important concept of crystal field theory. We do not discuss the effect of the crystal field in the splitting of orbital degeneracy. We exclusively discuss its effect on the nature of exchange energy *J* involved in the coupling of first-nearest spin neighbors. We show that, starting from an isotropic (Heisenberg) exchange coupling when there is no crystal field, the appearance of a crystal field induces an anisotropy of exchange coupling thus leading to a *z-z* (Ising-like) coupling or a *x-y* one. For the sake of simplicity, we do not consider the other cases. Thus, we only discuss the effects of a weak crystal field magnitude (3*d* ions) compared to a stronger (4*d* ions) and even stronger one (5*d* ions).

In a final step, without a crystal field (isotropic couplings) or with a crystal field (anisotropic couplings), we are then able to write the corresponding Hamiltonian exchange as a spin-spin exchange.

## 2. Microscopic Mechanisms Involved in Superexchange

### 2.1. Basic Physical Considerations

2.1.1. Generalities and Hund's Rules

When considering lattices composed of transition ions, there is a more or less important effect coming from the electrostatic potential due to the ionic environment: this effect is well known as that of the "crystal field". It, finally, describes the coulombic interactions inside the ionic cage in which the transition ion is inserted. When considering the electronic shells responsible for the magnetic properties, *d* orbitals are the most external shells of the ion. As a result, *d* orbitals are more "sensitive" to an external potential i.e., the crystal field.

*Thus, for 3d orbitals, we deal with a (more or less) weak crystal field. Oppositely the crystal field is strong when dealing with 4d and 5d orbitals.* This can be simply explained by the fact that, when passing from a 3*d* ion to a 5*d* one, the mean radius of the ion increases, whereas the same surrounding cage under consideration, in which the ion is inserted, shows similar dimensions. As a result, the magnitude of coulombic interactions also increases. These findings allow us to directly lead to the determination of adequate orbitals describing the ion's magnetic properties.

Hund's rules allow the determination of the orbital and spin momenta characterizing the ion's ground state. In fact, these rules try to explain how Coulomb repulsion and Pauli's exclusion principle must be considered simultaneously [10]. *If $S = S_{max}$, $L = L_{max}$ (for $S = S_{max}$) compatible with the exclusion principle are the respective values of the spin and orbit momenta along a z-axis of reference, the total momentum is $\mathcal{J} = L + S$. If the external electronic shell is at most half-filled ($L$ and $S$ are antiparallel) $\mathcal{J} = |L - S|$. If the shell is more than half-filled ($L$ and $S$ parallel) $\mathcal{J} = L + S$* (the unoccupied orbitals being considered as holes).

In Figure 1, we have reported the values of spin and orbital momenta characterizing a 3*d*, 4*d* or 5*d* external shell. The simplest situation is the electronic configuration $3d^1$ (ions $V^{4+}$ and $Ti^{3+}$), where a single orbital is filled. Thus, according to Hund's rules, the orbital momentum vanishes ($L = 0$, $S = 5/2$) at mid-filling of the 3*d* shell (case, $3d^5$, the ions, $Fe^{3+}$ and $Mn^{2+}$). As a result, the configuration $3d^6$ (the ion, $Fe^{2+}$, $L = 2$, $S = 2$) is equivalent to the case $3d^4$ (the ions $Mn^{3+}$ and $Cr^{2+}$). Similarly, the configuration $3d^1$ (ions $V^{4+}$ and $Ti^{3+}$, $L = 2$, $S = 1/2$) is equivalent to the case $3d^9$ (ion $Cu^{2+}$), and so on.

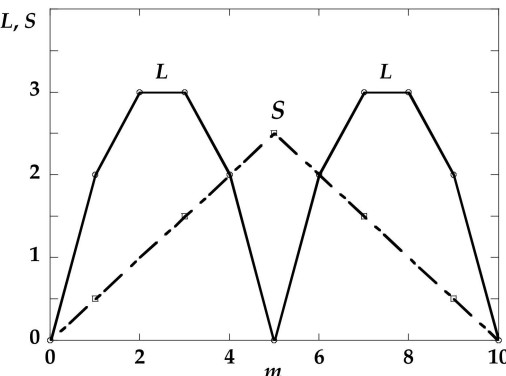

**Figure 1.** Values of spin $S$ and orbital $L$ momenta of an isolated transition ion, characterized by electronic shells $3d^m$, $4d^m$ or $5d^m$ ($1 \leq m \leq 10$).

It is well known that *for transition ions considered in the ground state, the orbital momentum is quenched.* As a result, *the magnetic properties of such a kind of ion are exclusively due to its spin momentum.* Thus, when considering the formal coupling of two spins, each one belonging to a magnetic center, one can speak about *direct exchange*. If these ions are isolated, we deal with an *isotropic (Heisenberg) coupling*. If a diamagnetic ligand, X, is inserted between two magnetic centers, A and B, the entity A–X–B also shows magnetic properties. In that case, we have an *indirect exchange* between A and B through X, also known as *superexchange*.

In this article, in a first step, we exclusively consider the "toy model" of two magnetic centers, A and B, i.e., two transition ions, $3d^1$, characterized by a single spin, $\frac{1}{2}$, and coupled through a diamagnetic ligand X *without a crystal field*. Then we introduce a crystal field contribution. As it is one of *the main sources of anisotropy for exchange, we shall discuss its influence on the nature of anisotropic couplings (z-z – Ising-type – or x-y couplings)*.

In a second step, we generalize the corresponding results in the case of $3d^m$ ions ($1 < m \leq 10$), i.e., ions such as $S > \frac{1}{2}$. In both cases, we consider a situation in which there is no crystal field and one for which a crystal field is introduced.

2.1.2. The First "Historical" Model Proposed by Anderson for Superexchange

In order to explain the magnetic properties of a wide class of materials, such as oxides and fluorides of transition elements and also including "natural" garnets (notably ferrites), Anderson proposed a model explaining the "indirect" exchange phenomenon arising between magnetic centers, in spite of the fact that they were surrounded by non-magnetic entities: this is the *superexchange* phenomenon [1–4]. According to Anderson [4], "*superexchange acquired its name because of the relatively large distances over which the exchange effect was often found to act between ions, radicals or molecules*".

However, he knew that superexchange never occurs in a classical ferromagnet, i.e., a medium in which the magnetic centers are characterized by a spin quantum number, *S*, such as $S \geq 5/2$, even if it is diluted inside a paramagnetic metal where there are long distances between the ferromagnetic and paramagnetic entities characterized by *d* orbitals. Why? *Taking into account this important experimental observation, Anderson concluded that the origin of this phenomenon was purely quantum because any classical interpretation failed.*

As a result, in a first step, Anderson made the following "reasonable" assumptions:

- The direct overlap of the involved wave functions characterizing the pair of magnetic sites A and B separated by the non-magnetic ligand X vanishes;
- The ligand wave function is weakly modified by the presence of magnetic ions;
- This modification confers a magnetic character that is the origin of the exchange interactions between the pair of magnetic ions through the non-magnetic ligand.

In a second step, Anderson considered the simplest modification for describing the *unavoidable exchange* between magnetic sites A and B through the non-magnetic ligand X: the transfer of one electron of the ligand X characterized by a full *s*- or *p*-external shell into the *d*-external shell of the magnetic ion A (or B) (*cf.* Figure 2). This assumption was based on the following results:

- Experimental measurements confirmed the transfer while examining the hyperfine interaction between the ligand nuclear spin and that of the magnetic ion;
- It has been graphically demonstrated that the ligand wave function is partially magnetic with the expected degrees;
- The electronic transfer of the up (or down) spin of ligand X to the empty left (or right) *d* orbital must remain ballistic, i.e., it conserves the spin so that it leads to an antiferromagnetic coupling;
- The comparison between experimental results obtained for a diluted or a concentrated sample of the same compound shows that the electronic transfer is weak so that the involved wave functions are weakly disturbed [19–25].

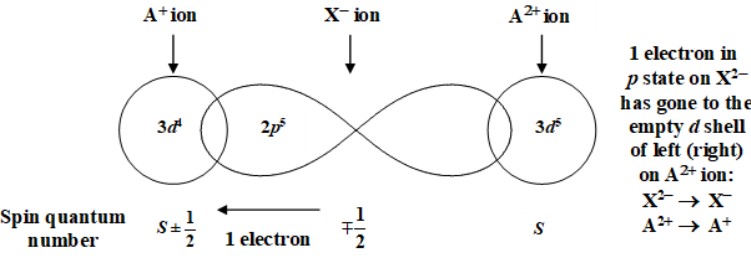

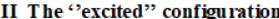

$$\psi(S_{\text{tot}}) = a\psi_{\text{ionic}}(S_{\text{tot}}) + b\psi_{\text{excited}}(S_{\text{tot}})$$

**Figure 2.** "Ground" and "excited" configurations in the original superexchange process for the sequence A–X–A; the electronic configuration of the valence shell has been added for each ion (in our case, A = Mn, X = O, for instance); due to the weak overlap, coefficients *a* and *b* are small. In the present case, the electronic transfer is from X to A but it can also occur from X to B, indifferently (here, B = A).

As a result, if $\psi_{\text{ionic}}(S_{\text{tot}})$ and $\psi_{\text{excited}}(S_{\text{tot}})$ are the respective wave functions of the starting ionic configuration and the excited one, the final state of the entity A–X–A is described via the global wave function $\psi(S_{\text{tot}}) = a\psi_{\text{ionic}}(S_{\text{tot}}) + b\psi_{\text{excited}}(S_{\text{tot}})$, where *a* and *b* are small.

Anderson has defined the electronic transfer by the integral $b_{mm'}(\tau) = \int dr \varphi_m^*(r-n) \times (T+V)\varphi_{m'}(r-(n+\tau))$, *m* and *m'* labeling the corresponding involved orbitals, with one electron per *d* orbital. $H = T + V$ is the electronic Hamiltonian, where *T* is the kinetic energy and *V*, the potential one. The variables **r** and **n** allow us to define the position of the concerned electron(s) and the **τ**'s represent the fundamental translations of the lattice. Due to lattice periodicity, the function $\varphi_m$ is a Wannier function. This contribution has been called *kinetic exchange* because, during the formation of the weak chemical bond between A and X, the antiferromagnetic coupling of electronic spins is characterized by a gain in kinetic energy. If *U* is the coulombic repulsion energy, Anderson has defined the corresponding exchange energy as $J_{m,m'}$ (kinetic) $= -2b_{m,m'}(\tau)^2/U$.

When the unpaired electrons belong to orthogonal orbitals spins are coupled ferromagnetically, the corresponding contribution is given by an integral of the type $J_{mm'}(\text{potential}) = \int dr dr' \varphi_m^*(r-n)\varphi_{m'}^*(r-n)(e^2/4\pi\varepsilon_0|r-r'|)\varphi_{m'}(r'-n)\varphi_m(r'-n)$. It is positive because the electrons "try" to avoid themselves through the *Fermi hole* that appears when spins are parallel (this hole becomes less repulsive when electrons are "closer"). Anderson called this contribution *potential exchange*.

The total exchange is finally $J_{mm'} = J_{mm'}(\text{potential}) + J_{mm'}(\text{kinetic})$, with the conventional writing of an exchange Hamiltonian, $-2J_{m,m'}s_A^m \cdot s_B^{m'}$ for a couple of *d* bands (*m*,*m'*), with one *d* band per ion [2,3,24].

Finally, one may search for the physical origin of the electronic transfer occurring between species A (or B) and X. It is correlated to the creation of a hole in the external electronic shell that the electron has left behind, so that we have to consider an electron-hole couple when describing the electronic transfer. At this stage, we may guess, from now on,

that coulombic interactions are going to play a fundamental role. Under these conditions, we deal with a *Mott-Wannier exciton* because it must occur not only on the whole dimer A–X–B but also over a long distance in a crystal composed of unit cells A–X–B.

*2.2. Starting Assumptions*

2.2.1. Assumption 1

In this section, we detail the general formalism allowing us to describe the superexchange phenomenon for a centrosymmetric entity AXB, where A and B are two magnetic sites and X is a closed-shell diamagnetic ligand (with one unpaired electron labeled 1, belonging to A, and electron 2 for the one transferred from X to A, for the fragment A–X). In our case, *as we deal with a linear entity,* A–X–B, *we have no Jahn–Teller effect* [26,27] and no polarization of the bonds A–X and X–B.

This model is built without the presence of a crystal field and exclusively concerns the case of $3d$ ions ($3d^1$ in a first step, then $3d^m$ ions, $1 < m \leq 10$). Regarding the compounds whose magnetic properties may be described, we have "natural" compounds, such as oxides, fluorides, and garnets (notably ferrites), that we call "Class I compounds". In these compounds, the average size of the ligand orbital ($p$ orbital) is close to that of the magnetic cation ($d$ orbital). With the appearance of new "synthetized" magnetic compounds at the beginning of the 1990s, the magnetic cations are well separated by more or less long organic ligands [15–18]. In other words, the average size of a cation orbital is plainly lower than that of a ligand. We call these compounds "Class II compounds".

From a physical point of view, for "Class I compounds", the coulombic interactions between magnetic cations are dominant but in the case of "Class II compounds", their influence becomes negligible. In both cases, we shall fix the common physical limit that allows the separation of both magnetic regimes.

As a result, the fragment A–X is characterized by the following two-particle Hamiltonian:

$$H_{AX} = T_1 + V_1 + T_2 + V_2 + \frac{e^2}{4\pi\varepsilon_0 r_{12}}, \quad r_{12} = |\boldsymbol{r}_1 - \boldsymbol{r}_2|. \tag{1}$$

$T_i = -\hbar^2 \nabla_i^2 / 2m_e$ is the electronic kinetic energy operator, where $m_e$ is the electron mass. The potential operators $V_1 = V(\boldsymbol{r}_1)$ and $V_2 = V(\boldsymbol{r}_2)$ include all the nucleus and extra electron contributions to the Coulomb field acting on electrons 1 and 2, involved in the σ bond of the fragment A–X. The unique electron 1 comes from the $3d^1$ external shell of the magnetic ion A, and electron 2 comes from the full external electronic shell of ligand X. As a result, we work within the framework of the *Hartree–Fock approximation*, i.e., the action of extra electrons over electrons 1 and 2 is taken into account through a *mean-field approximation*.

A similar Hamiltonian $H_{XB}$ may be written for the fragment X–B (with $H_{AX} = H_{XB}$ when A = B). The electron coming from the external shell of X is labeled 3, and the one coming from B is labeled 4. All the physical quantities derived from the Hamiltonian $H_{AX}$ (respectively, $H_{XB}$) are labeled $Q_{AX}$ for part A–X (respectively, $Q_{XB}$ for part X–B).

2.2.2. Assumption 2

The potential operator $V(\boldsymbol{r})$ commutes with any symmetry operator $\mathcal{O}$ whose action is described by the properties of a double group $\mathcal{G}$ with the identity operation $\mathbb{1}$. As we deal with a two-electron problem on both sides of ligand X, separately, the two low-lying states of the bonds A–X and X–B are a spin-singlet and a spin-triplet. Indeed, if the spin wave function $\chi(s_1,s_2) = |s_1,s_2> = |s_1> \otimes |s_2> = |s_1^z> \otimes |s_2^z> = |s_1^z,s_2^z>$ describes the spin states, we have four possible pairings: $|\uparrow\uparrow>$, $|\uparrow\downarrow>$, $|\downarrow\uparrow>$ and $|\downarrow\downarrow>$. If $S = s_1 + s_2, S^z = s_1^z + s_2^z$ we have two classes of possibilities for writing the spin wave function:

- $|0, 0\rangle = \frac{1}{\sqrt{2}}(|\uparrow\downarrow\rangle - |\downarrow\uparrow\rangle)$, $S = 0$ (singlet state),

- $\left.\begin{array}{l} |1, 1\rangle = |\uparrow\uparrow\rangle \\ |1, 0\rangle = \frac{1}{\sqrt{2}}(|\uparrow\downarrow\rangle + |\downarrow\uparrow\rangle), \\ |1, -1\rangle = |\downarrow\downarrow\rangle \end{array}\right\}$　　　$S = 1$ (triplet state).

The singlet state is odd with respect to the interchange of $s_1$ and $s_2$, while the triplet state is even.

### 2.2.3. Assumption 3

If $\Phi_a(r_1)$, respectively, $\Phi_b(r_2)$ describes the eigenstate of the Hamiltonian $H_1 = T_1 = p_1^2/2m_e$, where $m_e$ is the electron mass, respectively, $H_2 = T_2 = p_2^2/2m_e$, the secular equation $\det(H - E\mathbb{1}) = 0$ (where $\mathbb{1}$ is the identity matrix and $H = T_1 + T_2 + U(r_1,r_2)$) can be solved (cf. Equation (1)). The solutions are spatially symmetric and antisymmetric wave functions, i.e., $\Phi_{S/A}(r_1, r_2) = (\Phi_a(r_1)\Phi_b(r_2) \pm \Phi_a(r_2)\Phi_b(r_1))/\sqrt{2}$, with $\langle\Phi_S(r_1,r_2)|\Phi_A(r_1,r_2)\rangle = 0$ by construction.

### 2.2.4. Assumption 4

As there is no spin-orbit coupling, the collective wave function describing the coupling of a pair of electrons is as appears in $\Psi(u_1,u_2) = \Phi(r_1,r_2)\chi(s_1,s_2)$, where $u_i = (r_i,s_i)$. When combining the respective parity properties of functions $\Phi(r_1,r_2)$ and $\chi(s_1,s_2)$ with Pauli's exclusion principle, we must have:

- $S = 0$ $\chi(s_1,s_2) = |s_1,s_2\rangle$ odd, $\Phi(r_1,r_2) = \Phi_S(r_1,r_2)$ even,
- $S = 1$ $\chi(s_1,s_2) = |s_1,s_2\rangle$ even, $\Phi(r_1,r_2) = \Phi_A(r_1,r_2)$ odd.

It means that a singlet state ($S = 0$) is a non-magnetic state (i.e., a purely diamagnetic state) whereas the triplet state ($S = 1$) is a magnetic one.

### 2.2.5. Assumption 5

If $\Gamma_{1/2}$ is the representation of a spin $\frac{1}{2}$, and if coupling a pair of these spins, we then have to consider the operation $\Gamma_{1/2} \otimes \Gamma_{1/2} = \Gamma_0 \oplus \Gamma_1$, where $\oplus$ is the direct sum symbol. $\Gamma_0$ and $\Gamma_1$ are the corresponding irreducible representatives (irrep) with dim $\Gamma_0 = 1$, dim $\Gamma_1 = 3$ and dim $(\Gamma_{1/2} \otimes \Gamma_{1/2}) = 4$.

### 2.2.6. Assumption 6

If considering the energy levels of the entity A-X-B they are sufficiently close in energy to be both populated at room temperature. In Figure 3 we have reported the radial behavior of $V(r)$ as well as that of energy for the AXB entity. The three corresponding atomic orbitals which are magnetic orbitals [28–30] are $\Phi_A$ and $\Phi_B$ centered on A and B, respectively, and $\Phi_X$ centered on the bridge, X. The corresponding states are $|A\rangle$, $|B\rangle$ and $|X\rangle$. $\Phi_A$, $\Phi_B$ and $\Phi_X$ are assumed to be real and are considered as starting (non-disturbed) wave functions i.e., free atomic wave functions that give a spatial description of each of the states $|A\rangle$, $|B\rangle$ or $|X\rangle$. *In the present case, $\Phi_A$ and $\Phi_B$ are cationic d-orbitals and $\Phi_X$ is an anionic (s or p) orbital.*

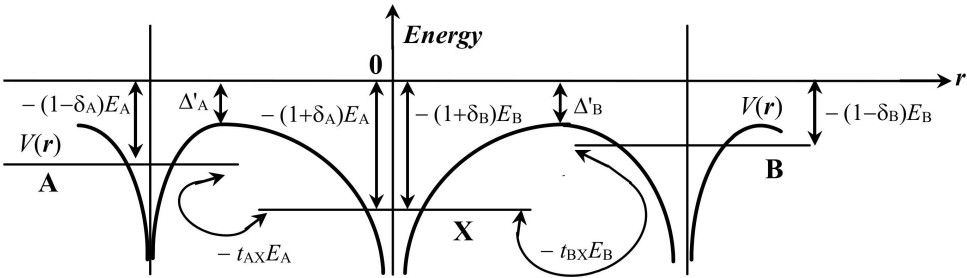

**Figure 3.** Radial behavior of energy for the centrosymmetric entity AXB in the general case of different magnetic sites (A ≠ B) on both sides of ligand X.

### 2.2.7. Assumption 7

*The following treatment is based on the symmetry group of the entity under consideration. The determination of this group is essential to find the relevant atomic orbitals which are permitted to combine in order to form "molecular" orbitals.* Let us recall that the electrons that are important are those belonging to the valence shell of A, B and X, respectively.

### 2.2.8. Assumption 8

We have the following properties based on the symmetry problem, notably on the fact that A and B are far apart and without interaction.

- The states are normalized but not orthogonal (except |A> and |B>):

$$<A \mid A> = 1, <B \mid B> = 1 , <X \mid X> = 1, <A \mid B> = <B \mid A> = 0. \tag{2}$$

As |A> and |B> are orthogonal (i.e., without overlap) there is no $\pi$ bond between A and B (but this could be taken into account in the present model). As noted after Equation (1), *only two unpaired electrons belonging, respectively, to cations A and B participate in the creation of a bond on each side of the central ligand X, the other electrons being considered as passive.* This is the *active-electron approximation,* which is plainly justified from an experimental point of view.

- The overlap between A and X on the one hand, and X and B on the other one, are defined as follows:

$$<A \mid X> = s_A, <X \mid B> = s_B, s_A > 0, s_B > 0. \tag{3}$$

In the particular case where A = B (degenerate case), then $s_A = s_B = s > 0$.

- Both magnetic sites A and B have a cationic energy level higher than the anionic one, as is generally the case for transition metal compounds; the energy difference between A and X levels (respectively, X and B levels) is $2\delta_A E_A$, with $E_A > 0$ (respectively, $2\delta_B E_B$, with $E_B > 0$), so that we have for the fragment A–X linked to X–B:

$$
\begin{aligned}
<A \mid (T_1 + V_1) \mid A> &= -(1 - \delta_A)E_A, \quad <B \mid (T_4 + V_4) \mid B> = -(1 - \delta_B)E_B, \\
<X \mid (T_2 + V_2) \mid X> &= -(1 + \delta_A)E_A, \quad E_A > 0, \quad <X \mid (T_3 + V_3) \mid X> = -(1 + \delta_B)E_B, \quad E_B > 0, \quad A \neq B, \\
<A \mid (T_1 + V_1) \mid A> &= <B \mid (T_4 + V_4) \mid B> = -(1 - \delta)E, \\
<X \mid (T_2 + V_2) \mid X> &= <X \mid (T_3 + V_3) \mid X> = -(1 + \delta)E, \quad E > 0, \quad A = B.
\end{aligned}
\tag{4}
$$

$\delta_A < 1$ for fragment A–X (respectively, $\delta_B$ for fragment X−B) but is not necessarily small. $\delta_i > 0$ (with $i$ = A or B) is a very common scenario, and $\delta_i < 0$ corresponds to the particular case of the dihydrogen molecule (with A = B, X being absent in that particular case).

A similar remark may be made for $\delta_{AB}$, which is defined by an analogy with respect to $\delta_A$ and characterizes the link between fragments A–X and X–B. In Figure 3, we also note that, due to the stabilization of AXB during the creation of the bonds A–X and X–B, we do have $-(1 + \delta_A)E_A = -(1 + \delta_B)E_B$. Of course, when A = B, $\delta_A = \delta_{AB} = \delta_B$.

- The transfer integrals between |A> and |X> on the one hand, and |X> and |B> on the other hand, are given by:

$$<A \mid (T_2 + V_2) \mid X> = -t_{AX}E_A, <X \mid (T_3 + V_3) \mid B> = -t_{XB}E_B, <A \mid (T_i + V_i) \mid B> = 0, i = 1,4. \tag{5}$$

If A = B, $t_{AX} = t_{XB} > 0$, but when A ≠ B, $t_{AX} \neq t_{XB} > 0$. The previous equation, stating that there is no transfer between A and B but exclusively between A and X or X and B, is a consequence of the condition imposed by Equation (2). However, as previously noted, *the case* $<A \mid (T_1 + V_1) \mid B> \neq 0$ *may be introduced in a more general model, without difficulty, notably by the bias of an $\pi$ orbital between* A *and* B.

- $s_A$, $t_{AX}$ and $t_{XB}$ are small compared to unity, and $t_{AX}$ or $t_{XB}$ is mainly related to the potential interaction between the anion and the cation, so that:

$$-t_{AX}E_A \approx -s_A\Delta'_A < -s_A(1 - \delta_A)\,E_A, \tag{6}$$

where $\Delta'_A$ is as defined in Figure 3. A similar relationship may be written for X−B, with $\Delta'_B$ also defined in Figure 3.

### 2.3. Expression of the Intermediate "Cationic" States

In this subsection, we exclusively consider the orbital (radial) part of the total wave function. As there is no direct overlap, i.e., no *direct transfer* between A and B, any exchange interaction between electrons belonging to the magnetic sites A and B automatically occurs through the anionic intermediate bridge, X. We use the starting rules suggested by Anderson [10]:

- The first rule consists of treating the extra electrons in terms of the simple one-electron Hartree–Fock functions;
- The second rule is treating them as excitations of a many-body system; this operation is achieved while keeping a constant value for the total spin involved, $S_{tot}$; this leads us to consider an ionic part for centers A and B and an excited one for the ligand X;
- As we deal with weak energies involved in the process of excitation, the orbital part $\psi(S_{tot})$ may be written as the following hybridization: $\psi(S_{tot}) = a\psi_{ionic}(S_{tot}) + b\psi_{excited}(S_{tot})$, where the coefficients $a$ and $b$ must remain small.

We diagonalize the one-particle Hamiltonian $T_i + V_i$ ($i$ = 1,2) in the reduced basis {|A>, |X>} for the fragment A–X (respectively, $T_i + V_i$ ($i$ = 3,4) in the reduced basis {|X>, |B>} for the fragment X−B, separately). The goal of such an operation is to obtain the new cationic (antibonding) normalized eigenstates |A> and |B>, such as:

$$|A> = (1 - \alpha_A)|A> + \beta_A|X>, |B> = \beta_B|X> + (1 - \alpha_B)|B> \tag{7}$$

where $\alpha_i$ and $\beta_i$ (with $i$ = A or B) are real numbers. As we deal with a weak chemical bond between A and X (respectively, between X and B), $\beta_i$ must remain small, and $\alpha_i$, close to unity. In addition, if using the normalization condition <A|A> = <B|B> = 1, as well as <A|X> = $s_A$, <X|B> = $s_B$ ($s_A > 0$, $s_B > 0$), the new normalization condition <A|A> = 1 and <B|B> = 1 leads to the following equation:

$$(1 - \alpha_i)^2 + 2\beta_i s_i(1 - \alpha_i) + \beta_i^2 - 1 = 0, i = \text{A or B}, \tag{8}$$

characterized by the solution $1 - \alpha_i = -\beta_i s_i + \sqrt{1 - \beta_i^2 + (\beta_i s_i)^2}$ (with $i$ = A or B). If defining the new direct overlap $S$, we derive owing to Equation (2):

$$S = <A|B> = <B|A> = \beta_B s_A(1 - \alpha_A) + \beta_A s_B(1 - \alpha_B) + \beta_A \beta_B > 0 \text{ A} \neq \text{B},$$
$$S = 2\beta s(1 - \alpha) + \beta^2 > 0, \text{ A} = \text{B}. \tag{9}$$

Setting:

$$S_A = 2\beta_A s_A(1 - \beta_A) + \beta_A^2 > 0, S_B = 2\beta_B s_B(1 - \beta_B) + \beta_B^2 > 0, \tag{10}$$

where $S_A$ and $S_B$ are functionals of the various overlaps corresponding to the bonds A–X and X−B, respectively, we have:

$$S = \frac{1}{2}\left(\frac{\beta_B}{\beta_A}S_A + \frac{\beta_A}{\beta_B}S_B\right) \tag{11}$$

and, by reporting in Equation (8), we have:

$$1 - \alpha_A = \pm\sqrt{1 - S_A}, 1 - \alpha_B = \pm\sqrt{1 - S_B}, \text{ A} \neq \text{B},$$
$$1 - \alpha = \pm\sqrt{1 - S}, \text{ A} = \text{B}. \tag{12}$$

From now on, the + sign is conventionally adopted. The new energy is:

$$E_A = <A|(T_1 + V_1|)A>, \; E_{AB} = <B|(T_1 + V_1)|B>, \; A \neq B,$$
$$E = E_A = E_{AB} = A|(T_1 + V_1)|A, \quad A = B. \tag{13}$$

Thus, if $A \neq B$ (general case), the system is non-degenerate ($E_A \neq |E_B$) but, if $A = B$, we are dealing with a degenerate state ($E = E_A = E_B$). Introducing the definition of $|A>$ and $|B>$ given by Equation (7), we may write, owing to Equations (4) and (5):

$$E_A = -[(1 - \alpha_A)^2(1 - \delta_A) + \beta_A^2(1 + \delta_A)]E_A - 2\beta_A(1 - \alpha_A)t_{AX}E_A,$$
$$E_{AB} = -[(1 - \alpha_B)^2(1 - \delta_{AB})E_B + \beta_B^2(1 + \delta_A)E_A] - 2\beta_B(1 - \alpha_B)t_{XB}E_B, A \neq B, \tag{14}$$
$$E = E_A = E_{AB} = -[(1 - \alpha)^2(1 - \delta) + \beta^2(1 + \delta)]E - 2\beta(1 - \alpha)tE, A = B.$$

The undetermined parameter, $\beta_i$ (with $i$ =A or B), may now be chosen so that $E_i$ is minimum, i.e., owing to Equation (14), the equation $\partial E_i/\partial \beta_I = 0$, after a few calculations, yields:

$$\beta_A = \pm(1 - \alpha_A) = \pm\sqrt{1 - S_A}, \; \beta_B = \pm(1 - \alpha_B) = \pm\sqrt{1 - S_B} \tag{15}$$

where $S_A$ and $S_B$ are given by Equation (10). Thus, $\beta_i$ and $1 - \alpha_i$ (with $i$ = A or B) may show the same sign or opposite signs: if $\beta_i > 0$ (respectively, $\beta_i < 0$) the state $|A>$ or $|B>$ will be represented by a spatially symmetric wave function (respectively, spatially antisymmetric). In addition, as $\beta_i$ is small, $1 - \alpha_i$ is also small, and $\alpha_i$ is close to unity, as expected. Then, using the particular value of $\beta_i$ given by Equation (15), the ground state energy is:

$$E_A = -2(1 - S_A)(1 \mp t_{AX})E_A,$$
$$E_{AB} = -(1 - S_B)[(1 - \delta_{AB})E_B + (1 + \delta_A)E_A \mp 2t_{XB}E_B], \; A \neq B, \tag{16}$$
$$E = E_A = E_{AB} = -2(1 - S)(1 \mp t)E_A, \; A = B$$

with $t = t_{AX} = t_{XB}$. The sign $\pm$ comes from that of $\beta$. We finally define the transfer integral $T_{AB}$ as:

$$T_{AB} = <A|(T_1 + V_1)|B> = <B|(T_1 + V_1)|A> . \tag{17}$$

Proceeding as for $S$ and $E$, we derive:

$$T_{AB} = -\beta_B(1 - \alpha_A)t_{AX}E_A - \beta_A(1 - \alpha_B)t_{XB}E_B - \beta_A\beta_B(1 + \delta_A)E_A, \; A \neq B,$$
$$T = -2\beta(1 - \alpha)tE - \beta^2(1 + \delta)E, \; A = B, \tag{18}$$

with $T = T_{AB} = T_{BA}$. When $A \neq B$, we always have $T_{AB} = T_{BA}$ on the condition that $(1 + \delta_A)E_A = (1 + \delta_B)E_B$ (cf. Figure 3). This illustrates the principle of indistinguishability of electrons: the electronic transfer may indifferently occur not only from X to A (case 1) but also from X to B (case 2), with the same physical effect. However, in both cases, we deal with the coupling of 2 electrons on the side of A or on the side of B.

In the particular case where there is no superexchange, we must have $S = <A|B> = <B|A> = 0$, so that $\alpha_A = \alpha_B = 0$, $\beta_A = \beta_B = 0$. From Equations (14) and (18), we derive, as expected, $E_A = -(1 - \delta_A)E_A$, $E_B = E_{AB} = -(1 - \delta_B)E_B$, $T_{AB} = 0$ (no transfer between A and B).

Then, using the particular value of $\beta_i$, (with $i$ = A or B) given by Equation (15), we have:

$$T_{AB} = -\sqrt{1 - S_A}\sqrt{1 - S_B}[(1 + \delta_A)E_A \pm t_{AX}E_A \pm t_{XB}E_B], \; A \neq B,$$
$$T = -(1 - S)(1 + \delta \pm 2t)E, \quad A = B, \tag{19}$$

where the + sign holds for $\beta_i > 0$ and − holds for $\beta_i < 0$. As $E_i > 0$, $1 - S_i > 0$, $1 + \delta_i > 0$ whatever the sign of $\delta_i$ $E_A \approx E_B > 0$, and $|t| << 1$, $T_{AB}$ (if $A \neq B$) or $T$ (if $A = B$) is *negative*. Thus, before constructing the collective states, it is clear that $S_A$, $E_A$ and $T_{AB}$ *(respectively, $S_B$, $E_B$ and $T_{BA}$) appear as the basic parameters of the bond* A–X *(respectively,* X−B*) and finally characterize the collective states of* AXB.

### 2.4. Construction of the Collective States

In this part, we wish to construct the wave function of the collective state describing the entity AXB. In the case of a fermionic assembly of $N$ particles, the collective wave function must appear as an antisymmetric combination of current terms $\Psi_{p_1}(u_1)\Psi_{p_2}(u_2)\ldots\Psi_{p_N}(u_N)$, where $u_i = (r_i, s_i)$ and $p_1, p_2, \ldots, p_N$ characterize the states occupied by these fermions. It is given by the following Slater determinant:

$$\Psi(u_1, u_2, \ldots, u_N) = \frac{1}{\sqrt{N!}}\begin{vmatrix} \Psi_{p_1}(u_1) & \Psi_{p_1}(u_2) & \ldots & \Psi_{p_1}(u_N) \\ \Psi_{p_2}(u_1) & \Psi_{p_2}(u_2) & \ldots & \Psi_{p_2}(u_N) \\ \ldots & \ldots & \ldots & \ldots \\ \Psi_{p_N}(u_1) & \Psi_{p_N}(u_2) & \ldots & \Psi_{p_N}(u_N) \end{vmatrix}. \tag{20}$$

Thus, the permutation of two particles is assumed by exchanging two columns, so that the determinant changes the sign. In addition, if any couple of numbers, $p_i$ and $p_j$, is as $p_i = p_j$, the determinant shows two identical lines and vanishes, in agreement with Pauli's principle, which states that two identic fermions cannot occupy the same state.

The "cationic" states $|A>$ and $|B>$ defined in the previous subsection may now give rise to four "cationic" spin-orbital states $|A,+>$, $|A,->$, $|B,+>$ and $|B,->$, from which we may construct, in a first step, four "molecular" states characterizing the entity AXB obeying a $\mathcal{G}$ symmetry group, thus conditioning the nature of the involved "molecular" orbitals.

Notably, owing to their behavior under the interchange of $|A,\sigma>$ and $|B,\sigma>$, $\sigma = \pm 1$, we call them "gerade" (unchanged) or "ungerade" (sign change), labeled $g$ and $u$, respectively:

$$\left| g, \sigma >= \frac{1}{\sqrt{2(1+S)}}(|A, \sigma > + |B, \sigma >), \right.$$
$$\left| u, \sigma >= \frac{1}{\sqrt{2(1-S)}}(|A, \sigma > - |B, \sigma >), \sigma = \pm 1 \right. \tag{21}$$

where the overlap $S$ is given by Equation (9). We here transpose the problem of sign change that was previously examined when interchanging 2 coupled fermions. Thus the *property "gerade" (unchanged) is analogous to the triplet character when coupling two electrons whereas the property "ungerade" (sign change) is analogous to the singlet character.*

The coefficients $[2(1 \pm S)]^{-1/2}$ are self-evident normalizing factors and $\sigma = \pm$ recalls the nature of the corresponding $\frac{1}{2}$ spin state ("up" or "down"). At this point, due to the orthogonality condition $<\mathcal{X},\sigma|\mathcal{Y},\sigma'> = \delta_{XY}\,\delta_{\sigma\sigma'}$ (with $\mathcal{X} = A$ or $B$), now, we must have $<X,\sigma|X,\sigma'> = \delta_{\sigma\sigma'}$, with $X = g$ or $u$. Then it is easily shown that the related energies are:

$$\begin{aligned} E_{Ag} &=< g\sigma|(T_1 + V_1)|g\sigma'> = \frac{E_A + 2T_{AB} + E_{AB}}{2(1+S)}\delta_{\sigma,\sigma'}, \\ E_{Au} &=< u\sigma|(T_1 + V_1)|u\sigma\prime> = \frac{E_A - 2T_{AB} + E_{AB}}{2(1-S)}\delta_{\sigma,\sigma'}, \ A \neq B, \\ E_{Ag} &=< g\sigma|(T_1 + V_1)|g\sigma'> = \frac{E+T}{1+S}\delta_{\sigma,\sigma'}, \\ E_{Au} &=< u\sigma|(T_1 + V_1)|u\sigma'> = \frac{E-T}{1-S}\delta_{\sigma,\sigma'}, \ A = B, \end{aligned} \tag{22}$$

where $E$ and $T$ are defined by Equations (13) and (17), respectively, and the difference of energy between the "gerade" and "ungerade" states is:

$$\begin{aligned} E_{Ag} - E_{Au} &= \frac{2T_{AB} - S(E_A + E_{AB})}{1 - S^2}, \ A \neq B, \\ E_{Ag} - E_{Au} &= 2\frac{T - SE}{1 - S^2}, \quad A = B, \end{aligned} \tag{23}$$

that is, as $\beta_i$, $s_i$ (with $i = A$ or $B$), $t$, $t_{AX}$, $t_{XB}$ (A = B or A $\neq$ B) and $S$ are small:

$$\begin{aligned} E_{Ag} - E_{Au} &\approx 2T_{AB} - S(E_A + E_{AB}) \approx 2T_{AB}, \ A \neq B, \\ E_{Ag} - E_{Au} &\approx 2(T - SE) \approx -2\beta^2(1 + \delta)E \approx 2T, \ A = B. \end{aligned} \tag{24}$$

Thus, $E_{Ag} - E_{Au}$ is independent of the sign of $\beta_i$, as expected, and remains very small. In the particular case where there is no superexchange, we recall that $S = 0$ and $T = 0$, so that $E_{Ag} = E_{Au}$.

From a quantum point of view, with two pairs of electrons and four available spin orbitals $|g,\pm\rangle$ and $|u,\pm\rangle$, $2^4$ determinantal collective states may then be built on each side of the fragment A–X–B. However, due to Pauli's exclusion principle, coupled to one of indistinguishability, when dealing with 4 fermions coupled in pairs (one per centers A and B, 2 for X with one electron possibly transferred to A, and one possibly transferred to B), the number of states reduces to $\begin{pmatrix} 4 \\ 2 \end{pmatrix} = 6$.

It means that the collective wave function must automatically be composed of 6 different states, each one itself being a $2 \times 2$ Slater determinant or a linear combination of these determinants.

As a result, let us label $|X_{S,S^z}\rangle$ the collective states. We have $X = U$ (ungerade) or $X = G$ (gerade) if referring to the symmetry of the orbital part with respect to the interchange of $|A\rangle$ and $|B\rangle$; $S$ and $S^z$ describe the total spin configuration. We define the collective basic state $|X,\sigma; Y,\sigma'\rangle$ as the corresponding Slater determinant:

$$|X, \sigma; Y, \sigma'\rangle = \frac{1}{\sqrt{2}} \begin{vmatrix} X(r_1)\sigma(s_1) & Y(r_1)\sigma'(s_1) \\ X(r_2)\sigma(s_2) & Y(r_2)\sigma'(s_2) \end{vmatrix}. \tag{25}$$

Clearly, it is immediately apparent that a combination of a *g*-type orbital and a *u*-type one allows us to obtain a *U*-type collective state, whereas combining two *g*- or two *u*-orbitals gives rise to a *G*-type orbital. As a result, we now may build the six following collective states:

$$\begin{aligned}
|U_{1,1}\rangle &= |u,+; g,+\rangle, \\
|U_{1,0}\rangle &= \tfrac{1}{\sqrt{2}}(|u,+; g,-\rangle + |u,-; g,+\rangle), \\
|U_{1,-1}\rangle &= |u,-; g,-\rangle, \\
|U_{0,0}\rangle &= \tfrac{1}{\sqrt{2}}(|u,+; g,-\rangle - |u,-; g,+\rangle), \\
|G_{0,0}^{g}\rangle &= |g,+; g,-\rangle, \\
|G_{0,0}^{u}\rangle &= |u,+; u,-\rangle.
\end{aligned} \tag{26}$$

It is very easily checked that these functions are orthogonal by construction. Their spin characters may be simply verified by applying to each of the previous collective states the convenient total spin operator. Then, introducing the expressions previously obtained for the molecular spin-orbital states (*cf* Equations (21) and (25)), we may express $|U_{11}\rangle, |U_{10}\rangle, |U_{1-1}\rangle, |U_{00}\rangle, |G_{0,0}^{g}\rangle$ and $|G_{0,0}^{u}\rangle$ on the basis of the following Slater determinants for the fragment A–X−B:

$$\begin{aligned}
|U_{1,1}\rangle &= \tfrac{1}{\sqrt{1-S^2}}|A,+; B,+\rangle, \\
|U_{1,0}\rangle &= \tfrac{1}{\sqrt{2(1-S^2)}}(|A,-; B,+\rangle + |A,+; B,-\rangle), \\
|U_{1-1}\rangle &= \tfrac{1}{\sqrt{1-S^2}}|A,-; B,-\rangle, \quad |U_{0,0}\rangle = \tfrac{1}{\sqrt{2(1-S^2)}}(|A,+; A,-\rangle - |B,+; B,-\rangle), \\
|G_{0,0}^{g}\rangle &= \tfrac{1}{2(1+S)}(|A,+; A,-\rangle + |A,+; B,-\rangle + |B,+; A,-\rangle + |B,+; B,-\rangle), \\
|G_{0,0}^{u}\rangle &= \tfrac{1}{2(1-S)}(|A,+; A,-\rangle - |A,+; B,-\rangle - |B,+; A,-\rangle + |B,+; B,-\rangle).
\end{aligned} \tag{27}$$

Concerning these latter states, it is useful to notice that they may be also expressed by means of the polar and non-polar, normalized (but non-strictly orthogonal) states $|G_{0,0}^{p}\rangle$ and $|G_{0,0}^{np}\rangle$, respectively:

$$\left|G_{0,0}^{g}>=\frac{\sqrt{1+S^2}}{\sqrt{2}(1+S)}\left(\left|G_{0,0}^{p}>+\right|G_{0,0}^{np}>\right),\right.$$

$$\left|G_{0,0}^{u}>=\frac{\sqrt{1+S^2}}{\sqrt{2}(1-S)}\left(\left|G_{0,0}^{p}>-\right|G_{0,0}^{np}>\right),\right. \tag{28}$$

with:

$$\left|G_{0,0}^{p}>=\frac{1}{\sqrt{2(1+S^2)}}(|A,+;A,->+|B,+;B,->),\right.$$

$$\left|G_{0,0}^{np}>=\frac{1}{\sqrt{2(1+S^2)}}(|A,+;B,->+|B,+;A,->).\right. \tag{29}$$

### 2.5. The Hamiltonian Matrix and Energy Spectrum

Now, we calculate the elements of the Hamiltonian matrix in the new basis $\{\,|\,G_{0,0}^{g}>,$ $|\,G_{0,0}^{u}>,\,|\,U_{0,0}>,\,|\,U_{1,1}>,\,|\,U_{1,0}>,\,|\,U_{1,-1}>\}$. The non-vanishing terms are those existing between states belonging to the same irreducible representation (irrep) of the orbital $\mathcal{G}$ and spin $\mathcal{R}$ symmetry groups so that the final group is $\mathcal{G}\otimes\mathcal{R}$. As a result, one may expect:

- Diagonal and off-diagonal terms between $|\,G_{0,0}^{g}>$ and $|\,G_{0,0}^{u}>$;
- Only diagonal terms for the states $|\,U_{S,S^z}>$ with $S=0$ ($S^z=0$) and $S=1$ ($S^z=0,\pm1$);
- All the diagonal terms of the states $|\,U_{1,S^z}>$ are equal because we deal with the irrep $\Gamma_1\otimes\Gamma_{3,u}$. Under these conditions, the Hamiltonian matrix is:

$$H=\begin{pmatrix} E_0^{Gg} & K & 0 & 0 & 0 & 0 \\ K & E_0^{Gu} & 0 & 0 & 0 & 0 \\ 0 & 0 & E_0^{U} & 0 & 0 & 0 \\ 0 & 0 & 0 & E_1^{U} & 0 & 0 \\ 0 & 0 & 0 & 0 & E_1^{U} & 0 \\ 0 & 0 & 0 & 0 & 0 & E_1^{U} \end{pmatrix} \tag{30}$$

with:

$$E_0^{Gg}=<G_{0,0}^{g}\left|H\right|G_{0,0}^{g}>,\ K=<G_{0,0}^{g}\left|H\right|G_{0,0}^{u}>=<G_{0,0}^{u}\left|H\right|G_{0,0}^{g}>,$$

$$E_0^{Gu}=<G_{0,0}^{u}\left|H\right|G_{0,0}^{u}>,E_0^{U}=<\ U_{0,0}|H|U_{0,0}>,$$

$$E_1^{U}=<\ U_{1,1}|H|U_{1,1}>=<\ U_{1,0}|H|U_{1,0}>=<\ U_{1,-1}|H|UU_{1,-1}>. \tag{31}$$

For the sake of simplicity, we restrict the results to the fragment $A_1-X-A_2$ (A = B). Then, we define the following quantities:

$$U_A=<A_1|<A_1|U_{1,2}|A_1>|A_1>,\ U_B=<A_2|<A_2|U_{1,2}|A_2>|A_2>=U_A=U,\ \text{A = B}$$

$$C=<A_1|<A_2|U_{1,2}|A_2>|A_1>, \tag{32}$$

$$\gamma_1=<A_1|<A_2|U_{1,2}|A_1>|A_2>,\ \gamma_2=<A_1|<A_1|U_{1,2}|A_1>|A_2>$$

with:

$$<W|<X|U_{1,2}|Y>|Z>=\int d\boldsymbol{r}_1 d\boldsymbol{r}_2\Phi_W^*(\boldsymbol{r}_1)\Phi_X^*(\boldsymbol{r}_2)\frac{e^2}{4\pi\varepsilon_0 r_{12}}\Phi_Y(\boldsymbol{r}_2)\Phi_Z(\boldsymbol{r}_1) \tag{33}$$

where $r_{12}=|\boldsymbol{r}_1-\boldsymbol{r}_2|$, the electron labeled 1 is coming from A, the one labeled 2, from X. The physical meaning of the parameters $U$, $C$, $\gamma_1$ and $\gamma_2$ is simply the following one:

- $U$ is the Coulomb energy for an electron pair occupying the same site;
- $C$ is the Coulomb energy for two electrons occupying neighboring sites;
- $\gamma_1$ is the Coulomb self-energy of the exchange charge distribution $-e\Phi_A(r)\Phi_B(r)$ and is, thus, referred to as the exchange integral;
- $\gamma_2$ appears as the Coulomb energy between the exchange charge distribution and an electron charge localized on one site. $\gamma_2$ is a transfer integral between two cationic orbitals, resulting from the effective coulombic potential created by the charge of another electron involved in the secular problem;
- When there is no superexchange, i.e., no exchange between A and B through X, we have $\gamma_1 \neq 0$ (the exchange charge distribution is restricted to the bond between A and X, X and B), $\gamma_2 = 0$ as there are no more cationic orbitals and $U \neq 0$, $C \neq 0$ (the Coulomb energy for two electrons is restricted to first neighboring sites: A and X or X and B).

When $A \neq B$, $U_A \neq U_B$. $C$, $\gamma_1$ and $\gamma_2$ conserve the same definition but not the same value as in the case where A = B.

Under these conditions, the matrix elements given by Equation (31) may easily be calculated:

$$E_0^{Gg} = 2E_{Ag} + \frac{U + C + 2\gamma_1 + 4\gamma_2}{2(1+S)^2}, E_0^{Gu} = 2E_{Au} + \frac{U + C + 2\gamma_1 - 4\gamma_2}{2(1-S)^2}, \quad (34)$$

$$K = \frac{U - C}{2(1 - S^2)}, \quad (35)$$

$$E_0^U = E_{Ag} + E_{Au} + \frac{U - \gamma_1}{1 - S^2}, E_1^U = E_{Ag} + E_{Au} + \frac{C - \gamma_1}{1 - S^2}, \text{ A = B}, \quad (36)$$

where $E_{Ag}$, $E_{Au}$ are given by Equation (22) for A = B or A $\neq$ B and $U$, $C$, $\gamma_1$ and $\gamma_2$ by Equation (32). When A $\neq$ B $U$ is replaced by $(U_A + U_B)/2$, the contribution of $C$, $\gamma_1$ and $\gamma_2$ is unchanged, though showing a different value than in the case where A = B. In addition, by diagonalizing the upper $2 \times 2$ matrix in Equation (30), we have the following eigenvalues:

$$E_{0,0}^{\pm} = \frac{1}{2}\left(E_0^{Gg} + E_0^{Gu} \pm \sqrt{\left(E_0^{Gg} - E_0^{Gu}\right)^2 + 4K^2}\right) \quad (37)$$

as well as the diagonal energy terms $E^p$ and $E^{np}$ for $|G_{0,0}^p\rangle$ and $|G_{0,0}^{np}\rangle$, respectively:

$$E^p = \frac{2(E + \mathcal{S}T) + U + \gamma_1}{1 + \mathcal{S}^2}, E^{np} = \frac{2(E + \mathcal{S}T) + C + \gamma_1}{1 + \mathcal{S}^2}. \quad (38)$$

### 3. Physical Interpretation

*3.1. Expression of $J_{m,m'}$*

As just seen in the previous subsection, when comparing the formal coupling of two electrons initially isolated, here we deal with:

- The states $|U_{S,S^z}\rangle$ with S = 1 ($S^z = 0, \pm1$) that are associated with a "triplet state", characterized by the eigenvalues $E_0^U$ and $E_1^U$ (three-times degenerated); and
- The states $|G_{0,0}^g\rangle$ and $|G_{0,0}^u\rangle$, with S = 0, are associated with a "singlet state", characterized by the eigenvalues $E_{0,0}^{\pm}$.

At this stage, when calling $E_{S,0}$ and $E_{T,0}$ the low-energy levels of the singlet and triplet spectra, respectively, it is worth recalling that the exchange energy may be defined according to two conventional writings:

- $J_{m,m'} = E_{S,0} - E_{T,0}$ with the corresponding Hamiltonian exchange $H^{ex} = -J_{m,m'}s_1 \cdot s_2$ (convention 1); in that case, $J < 0$ corresponds to an antiferromagnetic arrangement, with $E_{T,0} > E_{S,0}$, whereas $J > 0$ corresponds to a ferromagnetic one, with $E_{T,0} < E_{S,0}$, where $m$ and $m'$ are the name of $d$ bands located on each side of the ligand X.

- $J_{m,m'} = E_{T,0} - E_{S,0}$ with the corresponding Hamiltonian exchange $H^{\mathrm{ex}} = J_{m,m'}\boldsymbol{s}_1.\boldsymbol{s}_2$ (convention 2); in that case, $J > 0$ corresponds to an antiferromagnetic arrangement, with $E_{T,0} > E_{S,0}$, whereas $J < 0$ corresponds to a ferromagnetic one, also with $E_{T,0} < E_{S,0}$.

As often used by theoreticians, we here adopt convention 2. The spin-spin Hamiltonian $J\boldsymbol{s}_i.\boldsymbol{s}_j$, here $J = E_{T,0} - E_{S,0}$, is submitted according to a couple of conditions:

- $J << \Delta$
- $k_{\mathrm{B}}T << \Delta$

where $T$ is the absolute temperature, $k_{\mathrm{B}}$, the Boltzmann's constant, and $\Delta = E_{S,1} - E_{S,0}$; $E_{S,1}$ is the first excited-state energy in the singlet spectrum.

$E_{0,0}^{-}$ (respectively, $E_{1}^{U}$) is the lowest energy of the singlet (respectively, triplet) spectrum (*cf* Figure 4). Under these conditions, we have:

$$J_{m,m'} = E_{1}^{U} - E_{0,0}^{-} \tag{39}$$

where $E_{0,0}^{-}$ and $E_{1}^{U}$ are given by Equations (34)–(37). The resulting energy level scheme is reported in Figure 4. As a result, a general theoretical expression of $J_{m,m'}$ may be written for the fragment A–X−A:

$$J_{m,m'} = \frac{C - \gamma_1}{1 - S^2} - \frac{\left(1 + S^2\right)(U + C + 2\gamma_1) - 8S\gamma_2}{2\left(1 - S^2\right)^2}$$
$$+ \sqrt{\left(E_{\mathrm{A}g} - E_{\mathrm{A}u} - \frac{S(U + C + 2\gamma_1) - 2\left(1 + S^2\right)\gamma_2}{\left(1 - S^2\right)^2}\right)^2 + \left(\frac{U - C}{2}\right)^2}. \tag{40}$$

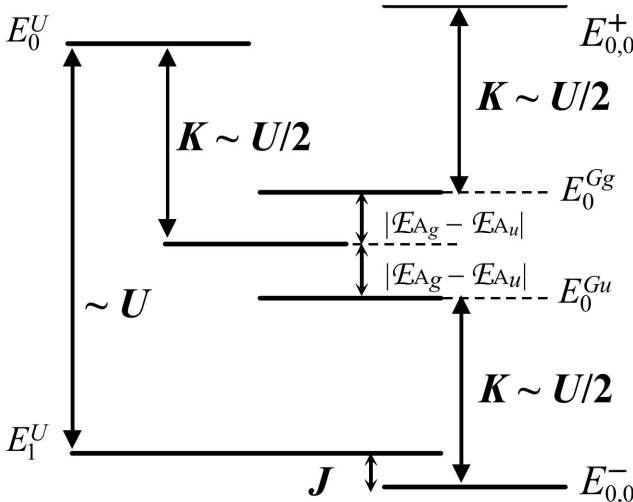

**Figure 4.** Energy level scheme for the AXB centrosymmetric system (with here A = B for the sake of simplicity) in the case of dominant coulombic interactions; the difference $|E_{\mathrm{A}g} - E_{\mathrm{A}u}|$ has been artificially zoomed for clarity. $J$ is given by Equations (39) and (43).

In the physical case of a small overlap $S$, the previous equation reduces to:

$$J_{m,m'} = -2\gamma_1 - \frac{U - C}{2} + \sqrt{\left(E_{\mathrm{A}g} - E_{\mathrm{A}u} - S(U + C + 2\gamma_1) + 2\gamma_2\right)^2 + \left(\frac{U - C}{2}\right)^2},$$
$$E_{\mathrm{A}g} - E_{\mathrm{A}u} \approx 2(T - \mathcal{S}E) \approx 2T, \ \mathcal{S} << 1 \tag{41}$$

where $E$, $T$, $U$, $C$, $\gamma_1$ and $\gamma_2$ are, respectively, given by Equations (16), (19) and (32). $E_{\mathrm{A}g} - E_{\mathrm{A}u} \approx 2T$ represents the electronic energy transfer between the magnetic cations and the non-magnetic ligand, in the case of weak overlap ($S << 1$).

Now, the physical criterion that allows us to describe compounds for which coulombic interactions are dominant naturally appears, i.e., $E_{Ag} - E_{Au} \approx 2T < U$, and compounds for which coulombic interactions become negligible, $E_{Ag} - E_{Au} \approx 2T > U$. However, we shall see that this first classification must also be slightly refined.

At this stage, it is worth noting that, in the absence of superexchange, there is no electronic transfer $T = 0$ and $> E_{Ag} - > E_{Au}|_{T=0} = 0$. Let $J_{m,m',0}$ be the corresponding exchange energy, always given by Equation (41), in which $T = 0$, $U \neq 0$, $C \neq 0$, $\gamma_1 \neq 0$ but $\gamma_2 = 0$. We retrieve from Equation (41) that $J_{m,m',0} = -2\gamma_1$, as expected. We just have an exchange between A and X on the one hand, and X and B on the other hand, without a connection between A and B.

In a previous paper [31], we evaluated the exchange energy magnitude between two atoms. We have shown that, in the case of a small overlap between A and X, and, if adapting to our conventional writing for Hamiltonian (here $H^{ex} = J\boldsymbol{s}_1.\boldsymbol{s}_2$), the exchange energy magnitude is $J \approx -(j - uS^2)$, where $u$ is the Hartree term (direct term) and $j$ the Fock term (exchange term), respectively, defined as:

$$u = \int dr_1 dr_2 |\Phi_A(r_1)|^2 \frac{e^2}{4\pi\varepsilon_0 r_{12}} |\Phi_X(r_2)|^2, j = \int dr_1 dr_2 \Phi_A^*(r_1)\Phi_X^*(r_2) \frac{e^2}{4\pi\varepsilon_0 r_{12}} \Phi_X(r_1)\Phi_A(r_2)$$

and similar expressions when A is replaced by B. If $S \ll 1$, on which we focus here, $J_{AX} = J_{XB} \approx j$. The magnitude of $J_{AX}$ (respectively, $J_{XB}$) reduces to the Fock (exchange) term, as expected, so that for the two bonds, A–X and X−B, considered separately, we have $J_0 \approx -2j$. If comparing with Equations (32) and (33), we have $j \approx \gamma_1$ and $u = U$ so that finally we retrieve:

$$J_{m,m',0} = J_0 \approx -2\gamma_1.$$

thus, validating Equation (41).

We derive the difference of exchange energy between the superexchange ($J$) and no superexchange ($J_{m,m',0} = J_0$):

$$J_{m,m'} - J_{m,m',0} = -\frac{U-C}{2} + \sqrt{\left(E_{Ag} - E_{Au} - S(U + C + 2\gamma_1) + 2\gamma_2\right)^2 + \left(\frac{U-C}{2}\right)^2} > 0,$$
$$S \ll 1.$$

However, it is worth writing $J_{m,m'}$ for the most commonly encountered physical case. Indeed, from the definitions of $\gamma_2$, $C$, $\gamma_1$ and $U$, given by Equation (32), we have the following classification:

$$\gamma_2 \ll C \approx \gamma_1 \ll U \tag{42}$$

where the corresponding "physical" values range from tenths of an eV to a few eV. In addition, from a mathematical point of view, this classification is also due to the respective values of $r_1$ and $r_2$ (cf. Equations (32) and (33)) appearing in the arguments of the functions, giving the spatial behavior of the involved atomic orbitals.

Under these conditions, i.e., in the case of weak overlap, $S \ll 1$ on which we focus:

$$J_{m,m'} - J_{m,m',0} = -\frac{U-C}{2} + \sqrt{\left(E_{Ag} - E_{Au}\right)^2 + \left(\frac{U-C}{2}\right)^2} > 0.$$

As a result, the physical discussion exclusively concerns the classification of the coulombic terms $U$, $C$ and $\gamma_1$ with respect to the difference $|E_{Ag} - E_{Au}|$, which is very small, $\gamma_2$ being strongly negligible. Thus, two kinds of situations may occur: $|E_{Ag} - E_{Au}| \approx 2|T| \ll U$ or $|E_{Ag} - E_{Au}| \approx 2|T| \gg U$.

As a result, we derive the universal relationships if $S << 1$:

$$J_{m,m'} - J_{m,m',0} = 2\left(\frac{|E_{Ag}-E_{Au}|}{U-C}\right)^2 \approx \frac{8T^2}{(U-C)^2} >> 1, \; |E_{Ag} - E_{Au}| \approx 2|T| << U,$$

$$J_{m,m'} - J_{m,m',0} = |E_{Ag} - E_{Au}|\left(1 - \frac{U-C}{2|E_{Ag}-E_{Au}|}\right) \approx 2|T|\left(1 - \frac{U-C}{4T}\right) \approx 2|T|,$$

$$|E_{Ag} - E_{Au}| \approx 2|T| >> U.$$

We conclude that it is the electronic transfer that enforces the superexchange.

- **Case 1:** $|E_{Ag} - E_{Au}| \approx 2|T| << U$ (see Figure 4).

The coulombic interactions, $U$, dominate the term $|E_{Ag} - E_{Au}| \approx 2|T|$, which represents the total electronic energy transfer between A and X, X and B. We are dealing with Class I compounds. For such materials, the size of the X-ligand orbital (*s*- or *p*-like) has the same order of magnitude as that of the A(B)-magnetic cation (*d*-like). We then have:

$$J_{m,m',1} \approx -2\gamma_1 + \frac{8T^2}{(U-C)^2} < 0 \tag{43}$$

$J_{m,m',1}$ may take values of order, varying between one eV to tenths of eV. We deal with ferromagnetic couplings. Thus, the coulombic interaction favors a ferromagnetic coupling. This is due to a subtle mechanism that is a consequence of the first Hund's rule. This aspect is detailed in Section 3.2.1.

- **Case 2:** $|E_{Ag} - E_{Au}| \approx 2|T| >> U$ (see Figure 5).

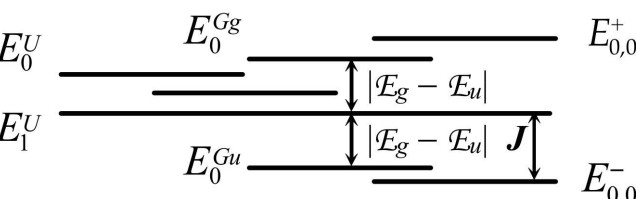

**Figure 5.** Energy level scheme for the AXB centrosymmetric system (with, here, A = B for the sake of simplicity) when Coulomb interactions are negligible (molecular orbital model). $J$ is given by Equations (39) and (44).

The contribution of coulombic interactions is now negligible vs. the small term $E_{Ag} - E_{Au} \approx 2T$, which represents the electronic transfer energy. When $E_{Ag} - E_{Au} \approx 2T >> 2S(U + C + 2\gamma_1)$, we then have:

$$J_{m,m',2} \approx |E_{Ag} - E_{Au}|\left(1 - \frac{U-C}{2|E_{Ag}-E_{Au}|}\right) \approx 2|T|\left(1 - \frac{U-C}{4T}\right) \approx 2|T| > 0 \tag{44}$$

$J_{m,m',2}$ shows a small value in eV, as suggested by Anderson's model. We are dealing with antiferromagnetic couplings. This mechanism is detailed in Section 3.2.2. because we now deal with a molecular orbital model. We may deal with Class I compounds ($S << 1$). However, this is always the case with Class II compounds because, due to the more or less important length of ligand X, the involved electrons show a very small coulombic interaction magnitude. $|E_{Ag} - E_{Au}| \approx 2|T + SE|$ is greater now, with $S < 1$.

When $|E_{Ag} - E_{Au}| << 2S(U + C + 2\gamma_1) | E_{Ag} - E_{Au} |$ is smaller and we again find case 1: the couplings are now ferromagnetic, but the corresponding value of $J$ is very small.

Here, there is an apparent contradiction with the experimental results. This would mean that there are no ferromagnets for Class II compounds characterized by a stronger

magnitude of *J*. In a previous paper ([10] and the references therein), we explained the strategy for obtaining molecular organic ferromagnets, as well as the main models illustrating the *charge transfer process*. All these approaches are characterized by the introduction of excited (anomalously polarized) configurations in the Hamiltonian basis, similar to the polar states introduced in our model. In fact, they do not drastically differ from that scheme for the energy-lowering term $4(\gamma_2 + T)^2/(U - C)$, involving the transfer integral *T*.

### 3.2. Physical Comments Regarding the Sign of J

3.2.1. Hund's First Rule

Let us recall that the states $|A>$ and $|B>$ defined by Equation (7) are *solutions of the same secular problem*. We have a weak overlap, $\mathcal{S}$. The eigenvalues of the $2 \times 2$ matrix appearing in the Hamiltonian expression given by Equations (30) and (34)–(37) may be written as:

$$E_{0,0}^{+} \approx 2E + U + \gamma_1 + \frac{4\gamma_2{}^2}{U - C}, E_{0,0}^{-} \approx 2E + C + \gamma_1 - \frac{4\gamma_2{}^2}{U - C}, \ S \ << \ 1 \qquad (45)$$

because $E_{Ag} + E_{Au} = 2E$, *E* being given by Equation (16). For an atom, $\gamma_2 = 0$ and, for a molecule or a polyatomic ion, $\gamma_2 \neq 0$. These eigenvalues must be compared to $E_0^U$ and $E_1^U$:

$$E_0^U = 2E + U - \gamma_1, E_1^U = 2E + C - \gamma_1, \ S \ << \ 1. \qquad (46)$$

The order of magnitude of the coulombic terms has been pointed out in Equation (42). The ground level is the triplet state $E_{T,0}$ and is stabilized by the factor $2\gamma_1$ (few eV for intra-atomic exchanges) with respect to the first excited level, $E_{S,0}$, even in the presence of orbital degeneracy (cf. Figure 4). The ferromagnetic interaction based on the Coulomb exchange integral $\gamma_1$ is called the *Heisenberg exchange*.

As the Pauli exclusion principle stipulates that two electrons cannot occupy the same position, when characterized by the same spin number, the corresponding wave function vanishes due to its property of antisymmetry. In other words, this function shows smooth variations, as requested for avoiding too large a kinetic energy.

Consequently, this keeps weak values when two electrons showing parallel spins are not too greatly separated. The probability density shows the same property, so that a hole appears in the up (down) spin distribution near one electron of the couple: this is *the Fermi hole*. In other words, the larger distance thus maintained between them allows us to explain that they are characterized by lower Coulomb energy compared to the scenario of antiparallel electronic spins.

3.2.2. Molecular Orbital Model

Here, the Coulomb interaction is small and, thus, plays a secondary role. The energy level scheme described in Figure 4 is replaced by that of Figure 5. Now, the ground state is given by one of the singlets $|g,+; g,->$ or $|u,+; u,->$, if taking into account their relative stabilities (*G*-type collective state). As a result, the first excited state is the triplet $|U_{1,S^z}>$, with $S^z = 0, \pm 1$.

This scenario occurs when the overlap between neighboring site orbitals is large (but $S < 1$), as it notably appears for C−C covalent bonds, for instance. Even in the case of orbital degeneracy, the low-lying state $|g,\sigma>$ or $|u,\sigma>$ is always available for the electronic pair, which may then couple in order to form a singlet or a triplet state. Finally, the first Hund's rule informs us that the triplet is more stable, as often occurs.

*3.3. Superexchange Hamiltonian for a 3d$^1$ ion. Generalization to an nd$^m$ ion ($3 \leq n \leq 5$, $1 \leq m \leq 10$)*

Anderson has shown that, for *d* bands labeled *m* and *m'*, characterizing each ion on both sides of ligand X, the corresponding Hamiltonian may be written:

$$H = \sum_{\substack{(m, m'), \\ n, \tau}} 2\frac{b_{mm'}(\tau)^2}{U} s_n^m . s_{n+\tau}^{m'} + C^{st} \tag{47}$$

where the $\tau$'s represent the fundamental translations of the lattice. $b_{m,m'}(\tau)$ is the transfer integral between *d* bands labeled *m* and *m'*; (*m,m'*) is the corresponding pair of involved bands; $U > 0$ is the Coulomb repulsion energy. Anderson has called this contribution the "kinetic" part of the exchange $J_{m,m'}$((kinetic). The "potential" part of the exchange, $J_{m,m'}$(potential), is defined at the end of Section 2.1.2, and the total exchange is given by $J_{m,m'} = J_{m,m'}$((kinetic) + $J_{m,m'}$(potential).

The formalism that describes the superexchange phenomenon, according to Anderson, exclusively reduces the Hamiltonian to the transfer integral, i.e., the kinetic contribution. In our model, we have seen that we have a more general formalism. We wish to show that the Hamiltonian may be written in the formula:

$$H = K s_1^m . s_2^{m'}, \; s_1^m . s_2^{m'} = \frac{1}{4}(P_T - 3P_S). \tag{48}$$

where $P_T$ and $P_S$ are projectors:

$$P_T = \frac{3}{4}\mathbb{1} + s_1 . s_2, \; P_S = \frac{1}{4}\mathbb{1} - s_1 . s_2 \tag{49}$$

acting in the triplet and singlet subspaces, respectively. *K* is a constant by which to determine values.

After a few algebraic calculations, it is easy to show that the projectors have the following properties:

$$P_T + P_S = \mathbb{1}, \; P_T \cdot P_S = P_S \cdot P_T = 0; \; P_T^t = P_T, P_S^t = P_S; P_T^\alpha = P_T, P_S^\alpha = P_S, \; \alpha > 0 \tag{50}$$

where $X^t$ is a transposed matrix, and $\mathbb{1}$ is the $4 \times 4$ identity matrix. Under these conditions, it is easy to show:

$$P_T - 3P_S = (P_T - P_S)(P_T - 3P_S) - 6P_S \tag{51}$$

so that:

$$H = \frac{K}{4}[(P_T - P_S)(P_T - 3P_S) - 6P_S]. \tag{52}$$

Let $|\Psi\rangle$ be the generic collective wave function and let us introduce the projector $|\Psi\rangle\langle\Psi| = 1$ on the basis vectors. We then have:

$$H|\Psi\rangle = \frac{K}{4}[(P_T - P_S)|\Psi\rangle\langle\Psi|(P_T - 3P_S) - 6P_S]|\Psi\rangle \tag{53}$$

with:

$$\langle\Psi|(P_T - 3P_S) = (P_T^t - 3P_S^t)|\Psi\rangle^* = (P_T - 3P_S)\langle\Psi| \tag{54}$$

due to Equation (50), and the fact that $|\Psi\rangle^* = \langle\Psi|$ because $|\Psi\rangle$ is real. Finally, using the fact that $\langle\Psi|\Psi\rangle = 1$, we may finally write:

$$H|\Psi\rangle = \frac{K}{4}[(P_T - P_S)|\Psi\rangle(P_T - 3P_S) - 6P_S|\Psi\rangle]. \tag{55}$$

According to the Hamilton principle, which states that in the true evolution of a physical system, described by the generalized coordinates $q(t)$, the action S must be minimal with

respect to $\boldsymbol{q}(t)$. The action S may be expressed with the Lagrangian system, $L\left(q(t), \overset{\bullet}{q}(t)\right)$, where $\overset{\bullet}{q}(t) = dq(t)/dt$ and, after an adequate Legendre transform, with the Hamiltonian system $H$. As a result, minimizing S means that, at the equilibrium, $H$ must be minimized. In gauge field theories, notably applied in particle physics, this process is called a *minimal coupling scheme*. Under these conditions, the equilibrium of any dynamic situation is described by the following Hamiltonian:

$$\min(H|\Psi\rangle) = \min\left(\frac{K}{4}[(P_T - P_S)|\Psi\rangle(P_T - 3P_S) - 6P_S|\Psi\rangle]\right) \tag{56}$$

where the operator min($x$) exclusively applies on a number x (for the Hamiltonian applied to a vector $|\Psi\rangle$, it concerns the result of the corresponding application). We then have:

$$\min\left(\frac{K}{4}(P_T - P_S)|\Psi\rangle\right) = \frac{1}{4}(E_{T,0}^{m,m'} - E_{S,0}^{m,m'})|\Psi\rangle = J_{m,m'}|\Psi\rangle,$$
$$\min(P_S|\Psi\rangle) = E_{S,0}^{m,m'}|\Psi\rangle \tag{57}$$

where $E_{T,0}^{m,m'}$ and $E_{S,0}^{m,m'}$ are the lowest energies of the triplet and singlet states, respectively, for $d$ bands $m$ and $m'$, characterizing each ion located on each side of the ligand, X. We recognize $J_{m,m'} = E_{T,0}^{m,m'} - E_{S,0}^{m,m'}$. As a result, at the equilibrium, we have:

$$H^{eq}|\Psi\rangle = \min(H|\Psi\rangle) = \frac{J_{m,m'}}{4}|\Psi\rangle(P_T - 3P_S) - \frac{3}{2}E_{S,0}^{m,m'}|\Psi\rangle, \ K = 1. \tag{58}$$

Due to Equations (54) and (49), we then have:

$$|\Psi\rangle(P_T - 3P_S) = \langle\Psi|^*(P_T - 3P_S) = (P_T^t - 3P_S^t)|\Psi\rangle = (P_T - 3P_S)|\Psi\rangle \tag{59}$$

so that finally, for an ion $3d^1$ characterized by bands $m$ and $m'$ involved on each side of ligand X:

$$H^{eq} = \frac{J_{m,m'}}{4}(P_T - 3P_S) - \frac{3}{2}E_{S,0}^{m,m'}1 = J_{m,m'}s_1^m s_2^{m'} - \frac{3}{2}E_{S,0}^{m,m'}1 \ \ (\text{ion } 3d^1) \tag{60}$$

where $J_{m,m'}$ is given by Equation (40).

When dealing with several $d$ bands located on each side of ligand X, we suppose that these bands are independent, i.e., non-interacting between themselves through the coulombic processes. We simply have:

$$H = \sum_{(m,m')} \left( J_{m,m'}s_1^m s_2^{m'} - \frac{3}{2}E_{S,0}^{m,m'}1 \right) \ \ (\text{ion } nd^m, \ 3 \le n \le 5, \ 1 \le m \le 5). \tag{61}$$

Indeed, as seen on Figure 1, for ions $nd^1$ to $nd^5$, we have different half-filled electronic shells, so that the value of the couple ($m,m'$) varies between 1 and 5. If this value is greater than 5, only the half-filled shells intervene in the calculation of $J_{m,m'}$ so that an ion of $nd^6$ ($L = 2, S = 2$), $nd^7$ ($L = 3, S = 3/2$), $nd^8$ ($L = 3, S = 1$) and $nd^9$ ($L = 2, S = 1/2$) is treated as an ion $nd^4$, $nd^3$, $nd^2$, and $nd^1$, respectively.

*3.4. Introduction of Crystal Field Theory*

3.4.1. Expression of $J_{m,m'}$; Physical Discussion of the Crystal Field Effect

The crystal field theory was notably introduced by Van Vleck in the 1930s [32]. When considering the ions of the cage in which each magnetic cation is inserted, we observe that each ion, *k,* can be described by a unique charge, $q_k$, due to the fact that there are very weak overlaps between first-nearest neighbors. We do not discuss in this paper the symmetry properties of the cage and the correlative simplifications that may occur.

In this subsection, we will only discuss the effect of the crystal field on the nature of exchange energy, *J*, involved in the coupling of first-nearest spin neighbors. We show that, starting from an isotropic (Heisenberg) exchange coupling when there is no crystal field, the appearance of a crystal field induces an anisotropy of exchange coupling, thus leading to a *z-z* (Ising-like) coupling or an *x-y* one. For the sake of simplicity, we do not consider the other cases, but the corresponding problem is solvable.

Thus, we only discuss the effects of a weak crystal field magnitude (3*d* ions) to a stronger (4*d* ions) and then stronger one (5*d* ions). Let us recall that this effect is due to the fact that the mean radius of the transition ion increases when passing from 3*d* ions to 5*d* ions, whereas the dimensions of the surrounding cage in which these ions are inserted roughly show the same size (we exclude the scenario of distortions that might occur).

However, we do not discuss the effect of the crystal field in the splitting of orbital degeneracy.

Let O be an arbitrary origin, $OX_1$, $OX_2$ and $OX_3$, three orthonormal axes and $r = (x_1, x_2, x_3)$ the Cartesian coordinates of any point M bearing an electron (with the usual correspondence, $x_1 = x$, $x_2 = y$, $x_3 = z$). As a result, the potential describing the crystal field effect may be written as:

$$V^{\mathrm{CF}}(r) = -\sum_k \frac{q_k e}{4\pi\varepsilon_0 |R_k - r|}, \ q_k e > 0 \text{ or } q_k e < 0 \tag{62}$$

where $R_k = (X_1, X_2, X_3)$ is the position vector associated with each ion *k* of the surrounding cage ($R_k = C^{\mathrm{st}}$). Considering the two cases, $q_k e > 0$ or $q_k e < 0$, means that we envisage the possibility not only of an electrically neutral entity constituted by the cage and the inserted cation A or B but also that the set cage + cation may show a positive or a negative electrical charge. Finally, we may write the full electrostatic contributions of the crystal field for the fragment A–X (electron 1 coming from cation A and electron 2 coming from anion X):

$$V^{\mathrm{CF}}_{\mathrm{A\text{-}X}}(r) = -\sum_k \frac{q_k e}{4\pi\varepsilon_0} \left( \frac{1}{|R_k - r_1|} + \frac{1}{|R_k - r_2|} \right)$$

and, for the fragment X−B (electron 3 coming from anion X and electron 4 coming from cation B):

$$V^{\mathrm{CF}}_{\mathrm{X\text{-}B}}(r) = -\sum_k \frac{q_k e}{4\pi\varepsilon_0} \left( \frac{1}{|R_k - r_3|} + \frac{1}{|R_k - r_4|} \right)$$

for similar insertion cages (with A = B or, exceptionally, for A $\neq$ B). Note that it is possible to consider different insertion cages, i.e., cages showing different geometrical properties. In this respect, we must add that this reasoning is valid for Class I compounds characterized by a ligand orbital length of the same order as cationic orbital length. However, for Class II compounds characterized by a ligand orbital length that is plainly greater than the cationic one, it has no physical sense by which to consider crystal field effects. The latter may only come from the electrostatic environment of the cation itself, which may be inserted within a crystallographic site.

As a result, the full contribution of the crystal field for the fragment A–X–B is:

$$U^{\mathrm{anis}} = V^{\mathrm{CF}}_{\mathrm{A\text{-}X}}(r) + V^{\mathrm{CF}}_{\mathrm{X\text{-}B}}(r) = -\sum_k \frac{q_k e}{4\pi\varepsilon_0} \sum_{i=1}^{4} \frac{1}{|R_k - r_i|}, \ A \neq B.$$

When A = B, on which we focus for the sake of simplicity, we have:

$$U^{\mathrm{anis}} = 2V^{\mathrm{CF}}_{\mathrm{A\text{-}X}}(r) = -2\sum_k \frac{q_k e}{4\pi\varepsilon_0} \sum_{i=1}^{2} \frac{1}{|R_k - r_i|}, \ A = B.$$

Under these conditions, only the coulombic contributions defined by Equation (32) and called $X^{\mathrm{iso}}$ in the absence of a crystal field are altered. We then have:

$$U^{\text{anis}} = U_{\text{A}}^{\text{anis}} = \langle A_1| \langle A_1| \left( U_{12} - 2\sum_{k}\sum_{i=1}^{2} \frac{q_k e}{4\pi\varepsilon_0 |R_k - r_i|} \right) |A_1\rangle |A_1\rangle = U_{\text{A}}^{\text{iso}} + U_{\text{A}}^{\text{CF}},$$

$$C^{\text{anis}} = \langle A_1| \langle A_2| \left( U_{12} - 2\sum_{k}\sum_{i=1}^{2} \frac{q_k e}{4\pi\varepsilon_0 |R_k - r_i|} \right) |A_2\rangle |A_1\rangle = C^{\text{iso}} + U_{\text{C}}^{\text{CF}},$$

$$\gamma_1^{\text{anis}} = \langle A_1| \langle A_2| \left( U_{12} - 2\sum_{k}\sum_{i=1}^{2} \frac{q_k e}{4\pi\varepsilon_0 |R_k - r_i|} \right) |A_1\rangle |A_2\rangle = \gamma_1^{\text{iso}} + U_{\gamma_1}^{\text{CF}},$$

$$\gamma_2^{\text{anis}} = \langle A_1| \langle A_1| \left( U_{12} - 2\sum_{k}\sum_{i=1}^{2} \frac{q_k e}{4\pi\varepsilon_0 |R_k - r_i|} \right) |A_1\rangle |A_2\rangle = \gamma_2^{\text{iso}} + U_{\gamma_2}^{\text{CF}}, \ \text{A} = \text{B}.$$

(63)

As previously noted, for Class II compounds characterized by long ligand chains, there is no crystal field and factor 2 must be replaced by unity in Equation (63). In addition, in Appendix A, we have shown that the following physical parameters $T$ and $V$, $E_\text{A}$ and $T$, $E_{\text{A}g}$ and $E_{\text{A}u}$, respectively, as given by Equations (1), (16), (18) and (22), the coulombic terms $U_A^{\text{iso}}$, $C^{\text{iso}}$, $\gamma_1^{\text{iso}}$ and $\gamma_2^{\text{iso}}$, given by Equation (32), and the coulombic contributions of crystal field $U_{\text{A}}^{\text{CF}}$, $U_{\text{C}}^{\text{CF}}$, $U_{\gamma_1}^{\text{CF}}$ and $U_{\gamma_2}^{\text{CF}}$ defined in Equation (63) can be labeled by the generic term $Y_k$ and decomposed as a sum of their respective Cartesian components along the axes $OX_1$, $OX_2$ and $OX_3$, called $Y_{k;x_i}$ (as a scalar is a zero-rank tensor):

$$Y_k = \sum_{i=1}^{3} Y_{k;x_i}, \ Y_{k;x_i} = \frac{Y_k}{3}$$

(64)

because, for each ion, $k$, we deal with a spherical symmetry (cf. Equations (A6)–(A10)). However, for a given similar $x_i$-component $Y_{k_{x_i}}$, components $Y_k$ characterized by different $k$ values are not equal (except when the site in which the cation is inserted shows particular symmetry properties). Due to the definitions of $U_{\text{A}}^{\text{CF}}$, $U_{\text{C}}^{\text{CF}}$, $U_{\gamma_1}^{\text{CF}}$ and $U_{\gamma_2}^{\text{CF}}$ (cf Equation (63)), we may give the same mathematical argument regarding the respective values of $r$ and $R_k$ appearing in the argument of the atomic orbitals (cf Equation (A12) in Appendix A). We have (cf. Equation (42)):

$$U_{\gamma_2}^{\text{CF}} << \ U_{\text{C}}^{\text{CF}} \approx \ U_{\gamma_1}^{\text{CF}} << U_{\text{A}}^{\text{CF}}$$

(65)

and a similar inequality with the Cartesian components, for a given $k$.

The Hamiltonian matrix, defined by Equation (30) in the absence of a crystal field, now appears as:

$$H = \sum_{i=1}^{3} H_{x_i}^{\text{anis}}, \ H_{x_i}^{\text{anis}} = \begin{pmatrix} E_{0,x_i}^{Gg,\text{anis}} & K_{x_i}^{\text{anis}} & 0 & 0 & 0 & 0 \\ K_{x_i}^{\text{anis}} & E_{0,x_i}^{Gu,\text{anis}} & 0 & 0 & 0 & 0 \\ 0 & 0 & E_{0,x_i}^{U,\text{anis}} & 0 & 0 & 0 \\ 0 & 0 & 0 & E_{1,x_i}^{U,\text{anis}} & 0 & 0 \\ 0 & 0 & 0 & 0 & E_{1,x_i}^{U,\text{anis}} & 0 \\ 0 & 0 & 0 & 0 & 0 & E_{1,x_i}^{U,\text{anis}} \end{pmatrix}$$

(66)

with:

$$E_{0,x_i}^{Gg,\text{anis}} = E_{0,x_i}^{Gg,\text{iso}} + \frac{U_{\text{A},x_i}^{\text{CF}} + U_{\text{C},x_i}^{\text{CF}} + 2U_{\gamma_1,x_i}^{\text{CF}} + 4U_{\gamma_2,x_i}^{\text{CF}}}{2(1+S)^2},$$

$$E_{0,x_i}^{Gu,\text{anis}} = E_{0,x_i}^{Gu,\text{iso}} + \frac{U_{\text{A},x_i}^{\text{CF}} + U_{\text{C},x_i}^{\text{CF}} + 2U_{\gamma_1,x_i}^{\text{CF}} - 4U_{\gamma_2,x_i}^{\text{CF}}}{2(1-S)^2},$$

$$K_{x_i}^{\text{anis}} = K_{x_i}^{\text{iso}} + \frac{U_{\text{A},x_i}^{\text{CF}} - U_{\text{C},x_i}^{\text{CF}}}{2(1-S^2)},$$

$$E_{0,x_i}^{U,\text{anis}} = E_{0,x_i}^{U,\text{iso}} + \frac{U_{\text{A},x_i}^{\text{CF}} - U_{\gamma_1,x_i}^{\text{CF}}}{1-S^2}, \ E_{1,x_i}^{U,\text{anis}} = E_{1,x_i}^{U,\text{iso}} + \frac{U_{\text{C},x_i}^{\text{CF}} - U_{\gamma_1,x_i}^{\text{CF}}}{1-S^2}, \ \text{A} = \text{B},$$

(67)

where the corresponding isotropic component, for each of the quantities above, is given by Equation (34). Similar remarks, given after Equation (36) and concerning the scenario

where A $\neq$ B, remain valid here. If diagonalizing the upper $2 \times 2$ matrix in Equation (66), we have the following eigenvalues:

$$E_{0,0,x_i}^{\pm,\text{anis}} = \frac{1}{2}\left(E_{0,x_i}^{Gg,\text{anis}} + E_{0,x_i}^{Gu,\text{anis}} \pm \sqrt{\left(E_{0,x_i}^{Gg,\text{anis}} - E_{0,x_i}^{Gu,\text{anis}}\right)^2 + 4\left(K_{x_i}^{\text{anis}}\right)^2}\right). \tag{68}$$

As a result, for the fragment A–X–A, the general theoretical expression of $J$ may be written from Equation (40), where each physical quantity is replaced by the same one in the presence of a crystal field and given by Equation (67):

$$J_{m,m'} = \frac{C^{\text{anis}} - \gamma_1^{\text{anis}}}{1-S^2} - \frac{\left(1+S^2\right)\left(U^{\text{anis}}+C^{\text{anis}}+2\gamma_1^{\text{anis}}\right)-8S\gamma_2^{\text{anis}}}{2(1-S^2)^2}$$
$$+\sqrt{\left(E_{Ag}^{\text{anis}} - E_{Au}^{\text{anis}} - \frac{S\left(U^{\text{anis}}+C^{\text{anis}}+2\gamma_1^{\text{anis}}\right)-2(1+S^2)\gamma_2^{\text{anis}}}{(1-S^2)^2}\right)^2 + \left(\frac{U^{\text{anis}}-C^{\text{anis}}}{2}\right)^2}. \tag{69}$$

In the physical case of a small overlap, $S$, on which we focus, the previous equation reduces to:

$$J_{m,m'} = -2\gamma_1^{\text{anis}} - \frac{U^{\text{anis}}-C^{\text{anis}}}{2}$$
$$+\sqrt{\left(E_{Ag}^{\text{anis}} - E_{Au}^{\text{anis}} - S\left(U^{\text{anis}} + C^{\text{anis}} + 2\gamma_1^{\text{anis}}\right) + 2\gamma_2^{\text{anis}}\right)^2 + \left(\frac{U^{\text{anis}}-C^{\text{anis}}}{2}\right)^2}, \tag{70}$$
$$S \ll 1$$

where all the physical quantities can be expressed along the axes $OX_1$, $OX_2$ and $OX_3$, according to Equation (64).

As discussed in the scenario where there is no crystal field after Equation (41), using similar reasoning, it becomes easy to show that the difference of exchange energy between superexchange ($J_{m,m'}$) and no superexchange ($J_{m,m',0}$) is:

$$J_{m,m'} - J_{m,m',0} = -\frac{U^{\text{anis}}-C^{\text{anis}}}{2}$$
$$+\sqrt{\left(E_{Ag}^{\text{anis}} - E_{Au}^{\text{anis}} - S\left(U^{\text{anis}} + C^{\text{anis}} + 2\gamma_1^{\text{anis}}\right) + 2\gamma_2^{\text{anis}}\right)^2 + \left(\frac{U^{\text{anis}}-C^{\text{anis}}}{2}\right)^2} > 0, S \ll 1.$$

As in the absence of a crystal field, we have to compare the magnitude of the coulombic interactions $U^{\text{anis}}$ to the quantity $\left|E_{Ag}^{\text{anis}} - E_{Au}^{\text{anis}}\right| = \left|E_{Ag}^{\text{iso}} - E_{Au}^{\text{iso}}\right| = \left|E_{Ag} - E_{Au}\right| = 2|T|$.

As a result, we may also derive the universal relationships if $S \ll 1$:

$$J_{m,m'} - J_{m,m',0} = 2\left(\frac{|E_{Ag}-E_{Au}|}{U^{\text{anis}}-C^{\text{anis}}}\right)^2 \approx \frac{8T^2}{\left(U^{\text{anis}}-C^{\text{anis}}\right)^2} \gg 1,$$
$$\left|E_{Ag} - E_{Au}\right| \approx 2|T| \ll U^{\text{anis}},$$
$$J_{m,m'} - J_{m,m',0} = \left|E_{Ag} - E_{Au}\right|\left(1 - \frac{U^{\text{anis}}-C^{\text{anis}}}{2|E_{Ag}-E_{Au}|}\right) \approx 2|T|\left(1 - \frac{U^{\text{anis}}-C^{\text{anis}}}{4|T|}\right) \approx 2|T|,$$
$$\left|E_{Ag} - E_{Au}\right| \approx 2|T| \gg U^{\text{anis}}.$$

This means that the appearance of a crystal field also strengthens the superexchange.

- **Case 1:** $\left|E_{Ag}^{\text{anis}} - E_{Au}^{\text{anis}}\right| = \left|E_{Ag}^{\text{iso}} - E_{Au}^{\text{iso}}\right| \approx 2|T| \ll U^{\text{anis}}$.

The coulombic interactions, including those coming from the crystal field, dominate the term $\left|E_{Ag}^{\text{anis}} - E_{Au}^{\text{anis}}\right| = \left|E_{Ag}^{\text{iso}} - E_{Au}^{\text{iso}}\right| \approx 2|T|$. We then have:

$$J_{m,m',1}^{\text{anis}} \approx -2\gamma_1^{\text{anis}} \approx -2\left(\gamma_1^{\text{iso}} + U_{\gamma_1}^{\text{CF}}\right) \approx J_{m,m',1}^{\text{iso}} - 2U_{\gamma_1}^{\text{CF}}. \tag{71}$$

with:

$$J_{m,m',1}^{anis} = \sum_{i=1}^{3} J_{m,m',x_i,1}^{anis} = J_{m,m',1}^{xx} + J_{m,m',1}^{yy} + J_{m,m\prime,1}^{zz}, U_{\gamma_1,x}^{CF} \neq U_{\gamma_1,y}^{CF} \neq U_{\gamma_1,z}^{CF}. \quad (72)$$

As discussed, when there is no crystal field, we also deal here with Class I compounds. For such materials, the size of the X-ligand orbital (*s*- or *p*-like) has the same order of magnitude as that of A(B)-magnetic cation (*d*-like). However, we may also have Class II compounds (the X-ligand length is plainly greater than in the cation), thus imposing a weak crystal ferromagnetic contribution in absolute value (see below).

If $U_{\gamma_1}^{CF} > 0$ ($q_k e < 0$), as $\gamma_1^{iso} > 0$ and $J_{m,m',1}^{iso} = -2\gamma_1^{iso} < 0$ (cf. Equation (43)), we always have $J_{m,m',1}^{anis} < 0$ according to Equation (71), and all the coulombic interactions favor a ferromagnetic coupling. This effect is enhanced by the crystal field (CF) contribution, independently of its magnitude. This is also due to a subtle mechanism that is a consequence of the first Hund's rule, in spite of the presence of the crystal field. This aspect has been detailed in Section 3.2.1. This allows us now to justify the fact that Class I and Class II compounds enter this case.

If $U_{\gamma_1}^{CF} < 0$ ($q_k e > 0$), we have two possibilities, when examining Equation (71):

- $\gamma_1^{iso} >> -2U_{\gamma_1}^{CF} > 0$; we deal with a weak CF contribution (in the case of $3d^m$ ions); $J_{m,m',1}^{anis} \approx -2\gamma_1^{iso} < 0$; the surrounding cage is mainly characterized by an important geometrical size: we may deal with Class I compounds. This is also the case when using organic ligands whose long length may be adapted to the magnetic system that one wishes to build up [15–18]; this is a good way to obtain isotropic (Heisenberg) spin-spin couplings for Class II compounds; we always have ferromagnetic couplings, including in the particular case of $\gamma_1^{iso} \approx -2U_{\gamma_1}^{CF} > 0$ so that, finally, the ferromagnetic coupling is strongly enhanced and $J_{m,m',1}^{anis} \approx -4\gamma_1^{iso} \approx -4U_{\gamma_1}^{CF} < 0$ (in the case of ions $4d^m$ and $5d^m$);

- $\gamma_1^{iso} << -2U_{\gamma_1}^{CF} > 0$; we deal with a strong CF contribution (case of $5d^m$ ions); now we have $J_{m,m',1}^{anis} \approx -2U_{\gamma_1}^{CF} > 0$; antiferromagnetic couplings are favored and this only concerns Class I compounds.

Now we examine the influence of the crystal field over the initial isotropic character of exchange coupling. As previously announced at the beginning of the present subsection, we restrict our study to those cases where the axis $OX_3 = Oz$ is favored (with respect to $OX_1 = Ox$ and $OX_2 = Oy$) or the axes $OX_1 = Ox$ and $OX_2 = Oy$ are equally favored by the crystal field (with respect to $OX_3 = Oz$). Then, we have:

$$\begin{aligned} J_{m,m',1}^{anis} &\approx J_{m,m',1}^{zz} \text{ if } U_{\gamma_1,x}^{CF} \approx U_{\gamma_1,y}^{CF} << U_{\gamma_1,z}^{CF} (z\text{-}z \text{ coupling; here Ising coupling as } s = 1/2),\\ J_{m,m',1}^{anis} &\approx J_{m,m',1}^{xx} + J_{m,m',1}^{yy} \text{ if } U_{\gamma_1,x}^{CF} \approx U_{\gamma_1,y}^{CF} >> U_{\gamma_1,z}^{CF} (x\text{-}y \text{ coupling}). \end{aligned} \quad (73)$$

We can have $J_{m,m',1}^{anis} < 0$ or $J_{m,m',1}^{anis} > 0$. This is in the case of $4d$ and $5d$ ions.

- **Case 2:** $\left| E_{Ag}^{anis} - E_{Au}^{anis} \right| = \left| E_{Ag}^{iso} - E_{Au}^{iso} \right| \approx 2|T| >> U^{anis}$

The contribution of coulombic interactions is weak vs. the small term $| E_{Ag} - E_{Au} |$, which is insensitive to the crystal field by definition. This is in the case of $3d$ ions.

We have the same discussion as in the isotropic case, i.e., without a crystal field. When $\left| E_{Ag}^{iso} - E_{Au}^{iso} \right| >> 2S(U + C + 2\gamma_1)$, we have:

$$J_{m,m',2} \approx \left| E_{Ag}^{iso} - E_{Au}^{iso} \right| > 0. \quad (74)$$

We always deal with antiferromagnetic couplings independently of the crystal field. This mechanism is detailed in Section 3.2.2 because we now deal with a molecular orbital model.

When $\left| E_{Ag}^{iso} - E_{Au}^{iso} \right| \ll 2S(U^{anis} + C^{anis} + 2\gamma_1^{anis}) \left| E_{Ag}^{iso} - E_{Au}^{iso} \right|$ is smaller, we find again Case 1: the couplings are now ferromagnetic, but the corresponding value of $J$ is very small.

As discussed, in the absence of a crystal field we deal with Class I compounds ($S \ll 1$). However, this is always the case with Class II compounds because, due to a more or less important length of ligand X, the involved electrons show a very small coulombic interaction magnitude. $\left| E_{Ag}^{iso} - E_{Au}^{iso} \right| \approx 2|T + SE|$ is greater, now $S < 1$. Similar remarks may be made for molecular organic ferromagnets.

### 3.4.2. Expression of the Hamiltonian

We wish to show that the Hamiltonian may be written in the form of:

$$H = \sum_{(m,m')} \left( J_{m,m'}^{xx} s_1^{x,m} . s_2^{x,m'} + J_{m,m'}^{yy} s_1^{y,m} . s_2^{y,m'} + J_{m,m'}^{zz} s_1^{z,m} . s_2^{z,m'} \right). \tag{75}$$

By an analogy with the isotropic case, we define the following projectors:

$$s_1^{x,m} . s_2^{x,m'} + s_1^{y,m} . s_2^{y,m'} = \frac{1}{4}(P_T - 3P_S)_{xy}, \quad s_1^{z,m} . s_2^{z,m'} = \frac{1}{4}(P_T - 3P_S)_{zz}, \tag{76}$$

with the properties:

$$(P_T - 3P_S)_{xy} + (P_T - 3P_S)_{zz} = P_T - 3P_S, \quad (P_T - 3P_S)_{xy} . (P_T - 3P_S)_{zz} = 0 \tag{77}$$

where $P_T$ and $P_S$ are projectors (defined by Equation (49)) and are acting in the triplet and singlet subspaces, respectively. $(P_T - 3P_S)_{xy}$ is the projector acting in the subspace corresponding to the $x$-$y$ plane and $(P_T - 3P_S)_{zz}$ acts along the $z$-axis.

Similar reasoning performed for the isotropic case, i.e., without a crystal field (cf. Section 3.3) allows one to directly write in the presence of a crystal field:

$$H^{eq,xy} = \frac{J_{m,m'}^{xy}}{4}(P_T - 3P_S)_{xy} - \frac{3}{2}E_{S,0}^{xy;m,m'}\mathbb{1} = J_{m,m'}^{xy}\left( s_1^{x,m} s_2^{x,m'} + s_1^{y,m} s_2^{y,m'} \right) - \frac{3}{2}E_{S,0}^{xy;m,m'}\mathbb{1}$$

$$x\text{–}y \text{ plane favored by CF (ion } nd^1, \ 3 \leq n \leq 5) \tag{78}$$

and:

$$H^{eq,zz} = \frac{J_{m,m'}^{zz}}{4}(P_T - 3P_S)_{zz} - \frac{3}{2}E_{S,0}^{zz;m,m'}\mathbb{1} = J_{m,m'}^{zz} s_1^{z,m} s_2^{z,m'} - \frac{3}{2}E_{S,0}^{zz;m,m'}\mathbb{1}$$

$$z\text{-axis favored by CF}\left(\text{ion } nd^1, \ 3 \leq n \leq 5\right) \tag{79}$$

where $J_{m,m'}^{xy}$ and $J_{m,m'}^{zz}$ are given by Equations (71)–(74).

When dealing with several $d$ bands, located on each side of ligand X, we again suppose that these bands are independent, i.e., non-interacting between themselves through coulombic processes. We simply have:

$$H = \sum_{(m,m')} \left( J_{m,m'}^{xy}\left( s_1^{x,m} s_2^{x,m'} + s_1^{y,m} s_2^{y,m'} \right) - \frac{3}{2}E_{S,0}^{xy;m,m'}\mathbb{1} + J_{m,m'}^{zz} s_1^{z,m} s_2^{z,m'} - \frac{3}{2}E_{S,0}^{zz;m,m'}\mathbb{1} \right)$$

$$(\text{ion } nd^m, \ 3 \leq 5, \ 1 \leq 5). \tag{80}$$

## 4. Conclusions

In this article, we have developed a general model allowing us to describe the underlying microscopic mechanisms leading to superexchange in terms of the isolated centrosymmetric fragment A–X–B, where A and B are magnetic sites bearing $d$ orbitals and X is a diamagnetic ligand involving an $s$ or $p$ orbital. We have considered not only the degenerate case (A = B) but also the general one (A $\neq$ B). The orbitals describing the states | A> and

|B> do not overlap but the important scenario of overlap through a $\pi$ bond may also be taken into account.

The energy spectrum has been constructed. Notably, from the single-triplet splitting, a closed-form expression of the exchange energy $J$ has been expressed vs. key molecular integrals, for a given $3d^1$ ion. For the first time, this result has allowed us to predict the sign of $J$ and its magnitude: when coulombic interactions are dominant, our model follows Hund's rule and we explain why couplings are automatically ferromagnetic ($J < 0$); when coulombic interactions are no longer dominant, our model is equivalent to the molecular orbital one and couplings are always antiferromagnetic ($J > 0$), except in one particular case where couplings are ferromagnetic but with a small absolute value of $J$.

Owing to this general formalism, we have shown for the first time that the corresponding Hamiltonian may be written using the formula $Js_1.s_2$. The model has been easily generalized in the case of the $nd^m$ ion ($3 \leq n \leq 5$, $1 \leq m \leq 10$, orbital degeneracy).

Finally, we have introduced crystal field theory and we have shown how we can pass from an isotropic (Heisenberg) coupling (no crystal field, $3d$ ions) to a stronger ($4d$ ions) and stronger ($5d$ ions) contribution, which introduces an anisotropic exchange coupling. The nature of the spin arrangement has been determined, vs. the crystal field magnitude. We have shown that our formalism also allows us to write a spin-exchange Hamiltonian.

Important generalizations may be made. Notably, spin polarization effects may be introduced for the fragment A–X–B, as well as spin-orbit coupling. In particular, if this latter contribution remains small, the formalism may be slightly altered through a perturbation expansion, the zeroth-order orbitals being replaced by new magnetic orbitals, taking into account the spin-orbit perturbation.

This work has previously been conducted by Moriya, but he used Anderson's theory as a starting point [33,34]. As a result, one can say that the Dzialoshinskii coupling is a "close parent" of the superexchange coupling [35,36]. Moriya has notably determined the crystal symmetries that must be fulfilled for favoring the establishment of an antisymmetric spin coupling. More recently, Katsnelson et al. also used Anderson's theory and an exact perturbation expansion of the total energy of weak ferromagnets showing a canted spin structure, with the unique assumption of local Hubbard-type interactions. This allows the expression of the modulus of the Dzialoshinskii–Moriya vector, with a natural separation into spin and orbital contributions [37]. However, these models do not take into account all the key molecular integrals detailed in this paper or allow the expression of superexchange energy $J$ exactly.

The most important key point concerns the possible generalization of our formalism to any kind of molecule or polyatomic ion, opening the possibility of exact calculations of exchange couplings in real molecules such as polymers or biopolymers. It is a promising and important field of research in biology. Finally, the exact knowledge of the energy spectrum may allow others to interpret the experiences of photomagnetism that have recently been published, concerning ions Mo(IV) and W(IV) [38].

**Funding:** This research received no external funding.

**Institutional Review Board Statement:** Not applicable.

**Informed Consent Statement:** Not applicable.

**Conflicts of Interest:** The author declares no conflict of interest.

## Appendix A

Let O be an arbitrary origin, $OX_1$, $OX_2$ and $OX_3$ three orthonormal axes and ($x_1$,$x_2$,$x_3$) the coordinates of any point M bearing a mass $m_e$ and a charge $-e$. The kinetic energy $T$, defined by Equation (1), may be written as:

$$T = \frac{1}{2}m_e \sum_{i=1}^{3} \dot{x_i}^2 = \sum_{i=1}^{3} T_{x_i}, \ \dot{x_i} = \frac{dx_i}{dt}, \ \boldsymbol{r_j} = \mathbf{OM}_j = (x_{j;1}, x_{j;2}, x_{j;3}), \ j = 1,2 \quad \text{(A1)}$$

where the *i*th-space coordinate of species *rj* is labeled $x_{j;i}$ for the sake of simplicity. The potential *V*, also defined in Equation (1), represents the interaction of one electron with respect to the rest of the cation. If dealing with a spherical system, we have:

$$V = k \int \frac{dr}{4\pi\varepsilon_0 r} = k \ln r + C^{st}, r = \sqrt{x_1^2 + x_2^2 + x_3^2} \tag{A2}$$

where $k = -Qe$ is the electric charge of cation. We neglect the gravitational potential energy. As we deal with spherical symmetry, we have three equal contributions when expressing *V* along each axis OX$_i$, *i* = 1, 3. As a result, after the adequate integration of the first of Equation (A2), *V is invariant by the permutation of $x_1$, $x_2$ and $x_3$, and may be artificially written*:

$$V = \sum_{i=1}^{3} V_{x_i}, V_{x_i} = \frac{V}{3}. \tag{A3}$$

From Equation (33), the coulombic interactions may be defined by the generic form:

$$U_c = <W| <X|U_{1,2}|Y> |Z> = \int dr_1 dr_2 \Phi_W^*(r_1)\Phi_X^*(r_2)\frac{K}{4\pi\varepsilon_0 r_{12}}\Phi_Y(r_2)\Phi_Z(r_1) \tag{A4}$$

where $r_{12} = |r_1 - r_2|$, $K = e^2$ for $U_A$, $U_B$, $\gamma_1$ and $\gamma_2$ given by Equation (33); $dr_i$ represents the elementary volume in spherical coordinates i.e., $r_i^2 \sin\theta_i d\theta_i d\varphi_i$. In all cases, $K > 0$. $r_1$ and $r_2$ are the position vectors representing electron 1 from cation A, respectively, electron 2 from anion X, for instance. In a spherical coordinate system, we deal with the triplet $(r_j, \theta_j, \varphi_j)$, defined as follows:

$$\theta_j = \arccos\left(\frac{x_{j;3}}{r_j}\right), \varphi_j = \arctan\left(\frac{x_{j;2}}{x_{j;1}}\right), r_j = \sqrt{\sum_{i=1}^{3}(x_{j;i})^2}, j = 1, 2. \tag{A5}$$

$1/r_{12}$ may be expressed under the symmetric form with respect to the exchange of indices 1 and 2 [39]:

$$\frac{1}{r_{12}} = \frac{4\pi}{\sqrt{r_1^2 + r_2^2}}\sum_{l=0}^{+\infty}\left\{\sum_{n=l,l+2}^{+\infty}K_{n,l}\left(\frac{r_1 r_2}{r_1^2 + r_2^2}\right)^n\right\}\sum_{m=-l}^{+l}Y_{l,m}^*(\theta_1, \varphi_1)Y_{l,m}(\theta_2, \varphi_2),$$
$$K_{n,l} = \frac{(2n-1)!!}{(n-l)!!(l+n+1)!!}. \tag{A6}$$

In the previous equation, the notation of the current index of the *n*-summation means that *n* only takes *l*-values with an (*l* + 2)-step; $Y_{l,m}(\theta, \varphi)$ represents the well-known spherical harmonics [39].

As a result, we have:

$$U_c = \frac{K}{\varepsilon_0}\int dr_1 dr_2 f(r_1, r_2) \tag{A7}$$

with:

$$f(r_1, r_2) = \sum_{l=0}^{+\infty}\left\{\sum_{n=l,l+2}^{+\infty}K_{n,l}\frac{(r_1 r_2)^n}{(r_1^2 + r_2^2)^{n+1/2}}\right\}\sum_{m=-l}^{+l}Y_{l,m}^*(\theta_1, \varphi_1)Y_{l,m}(\theta_2, \varphi_2) \times$$
$$\times \Phi_W^*(r_1)\Phi_X^*(r_2)\Phi_Y(r_2)\Phi_Z(r_1). \tag{A8}$$

The integration over angular variables is easy: each integral is equal to unity with the selection rule $l = 0$, $m = 0$. As a result, we finally have the radial contribution:

$$U_c = \frac{K}{\varepsilon_0}\sum_{n=0,2}^{+\infty}K_{n,0}\int dr_1 dr_2 \frac{(r_1 r_2)^n}{(r_1^2 + r_2^2)^{n+1/2}}\Phi_W^*(r_1)\Phi_X^*(r_2)\Phi_Y(r_2)\Phi_Z(r_1). \tag{A9}$$

In the previous equation, $n$ only takes even values. This complicated integral may be uniquely calculated numerically. In addition, as we deal with the isotropic case of spherical coordinates, $U_c$ may be decomposed into 3 Cartesian components, $U_{c_{x_i}}$, so that:

$$U_c = \sum_{i=1}^{3} U_{c_{x_i}}, U_{c_{x_i}} = \frac{U_c}{3}. \tag{A10}$$

Thus, any coulombic interaction $U_c$ may be decomposed into a system of Cartesian coordinates along each axis $OX_i$, $i = 1,3$.

Now let us introduce the notion of *crystal field*. For the sake of simplicity, we restrict ourselves to the case of a unique cation A or B (respectively, anion X) inserted within a crystallographic site. The reasoning may be extended to all the ions involved in A–X–B. The potential describing the crystal field effect may be written as:

$$V^{CF}(\boldsymbol{r}) = -\sum_k \frac{q_k e}{4\pi\varepsilon_0|\boldsymbol{R}_k - \boldsymbol{r}|} \tag{A11}$$

where $\boldsymbol{R}_k = (X_1, X_2, X_3)$ is the position vector associated with each ion $k$ ($R_k = C^{st}$) and $\boldsymbol{r} = (x_1, x_2, x_3)$ is that of electron belonging to cation A (or B) or anion X. Under these conditions, the corresponding crystal field contribution may be written as:

$$U_c^{CF} = -\sum_k \Phi_X^*(\boldsymbol{R}_k)\Phi_Y(\boldsymbol{R}_k)\int d\boldsymbol{r}\,\Phi_W^*(\boldsymbol{r})\frac{q_k e}{4\pi\varepsilon_0|\boldsymbol{R}_k - \boldsymbol{r}|}\Phi_Z(\boldsymbol{r}). \tag{A12}$$

The expansion of $|\boldsymbol{R}_k - \boldsymbol{r}|^{-1}$ has been examined by Hutchings [40] in the assumption $r < R_k$, near an origin of cubic point symmetry corresponding to the three most-encountered situations (see Figure 2 in [40]):

- The charges composing the cage are located at the corners of an octahedron (sixfold coordination); in that case, $\boldsymbol{R}_k = (a,0,0)$, $(-a,0,0)$, $(0,a,0)$, $(0,-a,0)$, $(0,0,a)$ and $(0,0,-a)$; $R_k = a$ is the distance of each corner from the origin;
- The charges are located at the corners of a cube (eightfold coordination); $\boldsymbol{R}_k = (a,a,a)$, $(-a,a,a)$, $(a,-a,a)$, $(-a,-a,a)$, $(a,-a,-a)$, $(a,a,-a)$, $(-a,a,-a)$ and $(-a,-a,-a)$; $R_k = a\sqrt{3}$;
- The charges are located at the corners of a tetrahedron (with two tetrahedra per cube, fourfold coordination); $\boldsymbol{R}_k = (a,a,a)$, $(-a,-a,a)$, $(a,-a,-a)$, $(-a,a,-a)$ for tetrahedron 1 and $\boldsymbol{R}_k = (a,-a,a)$, $(-a,a,a)$, $(-a,-a,-a)$, $(a,a,-a)$ for tetrahedron 2; $R_k = a\sqrt{3}$.

In a first step, Hutchings has expanded $|\boldsymbol{R}_k - \boldsymbol{r}|^{-1}$ in Cartesian coordinates, with the assumption that $r/R_k < 1$ [40]. Under these conditions, this author obtains a perturbation series analogous to the famous dipolar expansion. He shows that $V^{CF}(\boldsymbol{r})$ may be written with Cartesian coordinates under the generic form:

$$V^{CF}(x_1, x_2, x_3) = A + V^{CF}_{x_1} + V^{CF}_{x_2} + V^{CF}_{x_3} + V_{mix}(x_1, x_2, x_3) \tag{A13}$$

$A$ is a constant composed of a product between a numerical factor and a ratio whose numerator is the coordination number, $q$, and the denominator is the distance of the cage charges from the origin. $V^{CF}_{x_1}$ (respectively, $V^{CF}_{x_2}$ and $V^{CF}_{x_3}$) are the polynomial of even powers of exclusive variables $x_1$ (respectively, $x_2$, $x_3$).

$V^{CF}_{x_1}$, $V^{CF}_{x_2}$ and $V^{CF}_{x_3}$ may be derived from each other through a permutation of variables $x_1$, $x_2$, and $x_3$ but, according to site symmetries, we may have not only $V^{CF}_{x_1} = V^{CF}_{x_2} = V^{CF}_{x_3}$ but also $V^{CF}_{x_1} \neq V^{CF}_{x_2} \neq V^{CF}_{x_3}$. $V_{mix}(x_1, x_2, x_3)$ is a polynomial characterized by terms composed of products separately involving two variables $x_1$ and $x_2$, $x_2$ and $x_3$, $x_3$ and $x_1$, showing even different powers, but it is impossible to separate variables $x_1$, $x_2$, and $x_3$ (if odd powers intervene in higher-order terms, the final result of integration vanishes). As a result, Equation (A13) may be rewritten as:

$$V^{\mathrm{CF}}(x_1, x_2, x_3) = A + V^{\mathrm{CF}}_{x_1}(x_1) + V^{\mathrm{CF}}_{x_2}(x_2) + V^{\mathrm{CF}}_{x_3}(x_3) \\ + \widetilde{V}_{x_1,x_2}(x_1, x_2) + \widetilde{V}_{x_2,x_3}(x_2, x_3) + \widetilde{V}_{x_3,x_2}(x_3, x_2). \tag{A14}$$

This is a quadratic form. It is then possible to express $V^{\mathrm{CF}}(x_1, x_2, x_3)$ in a new eigenbasis, under the following form:

$$V^{\mathrm{CF}}(\widetilde{x}_1, \widetilde{x}_2, \widetilde{x}_3) = V^{\mathrm{CF}}_{\widetilde{x}_1} + V^{\mathrm{CF}}_{\widetilde{x}_2} + V^{\mathrm{CF}}_{\widetilde{x}_3}. \tag{A15}$$

If inserting this expansion in Equation (A12) and integrating over variables $x_1$, $x_2$ and $x_3$, $U^{\mathrm{CF}}_c$ appears under the form:

$$U^{\mathrm{CF}}_{\mathrm{c}} = U^{\mathrm{CF}}_{\widetilde{x}_1} + U^{\mathrm{CF}}_{\widetilde{x}_2} + U^{\mathrm{CF}}_{\widetilde{x}_3}, \ U^{\mathrm{CF}}_{\widetilde{x}_1} \neq U^{\mathrm{CF}}_{\widetilde{x}_2} \neq U^{\mathrm{CF}}_{\widetilde{x}_3}. \tag{A16}$$

Hutchings has also calculated the matrix element $U_{\mathrm{c}}$ given by Equation (A12), in which $|\boldsymbol{R}_k - \boldsymbol{r}|^{-1}$ is expanded in terms of spherical harmonics. The matrix element $U_{\mathrm{c}}$ and correlated general rules of calculation may be found from this expansion, using vector coupling coefficients (see [41], p. 37). Hutchings recalls the rules for determining nonzero matrix elements, which follow from the Wigner–Eckart theorem (see [41], p. 75).

The conclusion derived by Hutchings is that, in a final step, $U_{\mathrm{c}}$ may be also expressed owing to Cartesian components. This method allows us to stop the expansion vs. spherical harmonics $Y_{l,m}(\theta, \varphi)$ for small values of $l$ and $m$, if using symmetries derived from that of the host site, whereas the direct calculation of the integral giving $U^{\mathrm{CF}}_{\mathrm{c}}$ is longer because it is exclusively ruled by the convergence of the series, giving $V^{\mathrm{CF}}(x_1, x_2, x_3)$ (cf. Equation (A9)).

As a result, *the anisotropic contributions of the crystal field appear as one of the main origins of the anisotropy of exchange.*

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
