# Peer review of "The Microscopic Mechanisms Involved in Superexchange"

_magnetochemistry, doi:10.3390/magnetochemistry8010006_

Round 1
Reviewer 1 Report
The paper deals with formulating a simplified theory of the superexchange interaction, analogous to the Mott dimer-based evaluation of the kinetic exchange. The approach includes a larger number of parameters than the original Mott dimer, in order to be applicable to the usual antiferromagnetic as well as ferromagnetic (double) exchange. Such a model is interesting and it seems desirable for determining the spin interactions in practice. However, the major ideas have been published by the author in Ref. [10]. The novelty of the present manuscript is mainly related to the inlusion of the crystal field effect on the spin Hamiltonian (an anisotropic exchange).
There are plenty of misprints in the manuscript. Except a huge number of equation-compilation mistakes (which could be the guilt of the submission system), I have noticed a misprint in line 859; the indices 3 and j should be exchanged in the first of Eq. (A5), I suppose. Much more importantly. in line 867, Eq. (A7) does not make sense since even the physical units on the left and right hand sizes are different, (maybe, a product of two summations should be considered...). Resulting further equations are thus questionable and the author's approach to the crystal field treatment may require reconsideration.
Reviewer 2 Report
The manuscript “The Microscopic Mechanisms Involved in Superexchange” by Jacques Curély reviews the classical model of superexchange with some modifications. The exchange interaction and the superexchange interaction are the less understood subjects in the Solid States Physics. I believe, the author’s view on the subject will be interesting for the research community and, therefore, I recommend the publication of the manuscript in the Journal of magnetochemistry as it is.
The manuscript is written in the classical style of the 50th. There are almost no calculations, graphs or explanations figures. There are many oversimplified schematics formals. That makes the manuscript to looks as an old classical paper. It is good. It could be helpful for a new researcher in the field.
I have some friendly comments for the author,
- For a review, it is nice to explain at least something about where the superexchange interaction mainly occurs (oxide, garnets). It is nice to explain the reason why the superexchange interaction does not occur in a classical ferromagnetic metal, even though in a ferromagnetic metal diluted with a paramagnetic metal, the distance between neighbor magnetic d- orbital might be long.
It is better to explain about the superexchange in viruses in introduction. Otherwise, it looks bad that the viruses pops-up in only conclusion without an obvious reason.
- It is nice to explain the reason why the superexchange interaction is so strong and why the superexchange interaction is not a simple sum of two exchange interactions between 3 neighbor orbitals (It is much stronger).
- It is nice at least to mention about the Dzyaloshinskii-Moriya exchange interaction, which is a close cousin of the superexchange interaction.
Author Response
Please see the list of correction in the attached file.

Reviewer 3 Report
The manuscript by Jacques Curely is a deep theoretical analysis of super-exchange coupling mechanism in dimers made by two transition metal ions, where the unpaired electron is in the unfilled 3d orbitals. The author achieves to express the exchange mechanism in terms of fundamental atomic overlap integrals and provides a physical interpretation. However, prior to acceptance, some major issues should be addressed.
Since Magnetochemistry is a journal where both experimental and theoretical results are reported, the paper should be understandable, at least in its general conclusions and structure, also by a non-theoretician. In this regard, the author should provide if possible some examples of compounds where its analysis could be applied. For example: for which compounds do we expect a low magnitude of coulombic term? How the formulas can be applied to different transition metal dimers? Also, theoretical results could be presented.
Another related point is the novelty of this new paper with respect to the previous one (Curély, J. Magnetic Orbitals and Mechanisms of Exchange II. Superexchange. Monatsh. Chem. 136, 1013–1036 (2005). https://doi.org/10.1007/s00706-005-0306-y) The structure and the content look very similar. The above-mentioned discussion could be a way to add new insights to the current manuscript and to stress the newly published results. As stated in the introduction, the new part of the paper is the physical interpretation. However, this is lacking.
Author Response
Please see the attached file containing the adequate responses to your questions.

Round 2
Reviewer 1 Report
The author has made a large effort improving the description compared to the previous version of the manuscript. However, despite of detailed explanation, the major point of my criticism still remains unchanged in principle. According to the current-version description, Eq. (A8) presents the scalar product of infinitesimal vectors, (however, the dot between vectors is omitted on the left hand side). Upon this is explanained in the present version, the physical units on the left- and right-hand sides of (A8) are the same. Unfortunately, being the scalar product, the left-hand side of (A8) is not the element of the double volume integral in Eq. (A4), whereas, the author makes such an substitution in Eq. (A4). In the consequence, the component of the interaction matrix, the righ-hand side of Eq. (A9) is of incorrect physical unit...
Reviewer 3 Report
The author has addressed all my raised issues. I recommend accepting the paper in its current form.
Author Response
PLease see attached file devoted to the reply to referee 1's last report.

Round 3
Reviewer 1 Report
In the present version, the most questionable equations are removed from Appendix A. It seems that, in terms of the dimensionality, everything is correct now provided the functions ΦX(r) in Eq. (A9) are equal to ΦX(r) in Eq. (A8) up to the multiplication by the radial variable "r" (the Jacobians are included into the integral). I suggest to write this down explicitely.
I still notice a misprint (?) in Eq. (A2) or an inconsistency with the description above it. I guess that, according to the text, the intergral in Eq. (A2) should be omitted since, in Eq. (1), there is just the Hamiltonian (not any integral of it).
In my opinion, upon the above details are improved, the manuscript was ready to be published.